# Poisson Variational Autoencoder

**Hadi Vafaii**[1]
vafaii@berkeley.edu

**Dekel Galor**[1]
galor@berkeley.edu

**Jacob L. Yates**[1]
yates@berkeley.edu

[1]UC Berkeley

## Abstract

Variational autoencoders (VAEs) employ Bayesian inference to interpret sensory inputs, mirroring processes that occur in primate vision across both ventral [1] and dorsal [2] pathways. Despite their success, traditional VAEs rely on continuous latent variables, which deviates sharply from the discrete nature of biological neurons. Here, we developed the Poisson VAE ($\mathcal{P}$-VAE), a novel architecture that combines principles of predictive coding with a VAE that encodes inputs into discrete spike counts. Combining Poisson-distributed latent variables with predictive coding introduces a metabolic cost term in the model loss function, suggesting a relationship with sparse coding which we verify empirically. Additionally, we analyze the geometry of learned representations, contrasting the $\mathcal{P}$-VAE to alternative VAE models. We find that the $\mathcal{P}$-VAE encodes its inputs in relatively higher dimensions, facilitating linear separability of categories in a downstream classification task with a much better ($5\times$) sample efficiency. Our work provides an interpretable computational framework to study brain-like sensory processing and paves the way for a deeper understanding of perception as an inferential process.

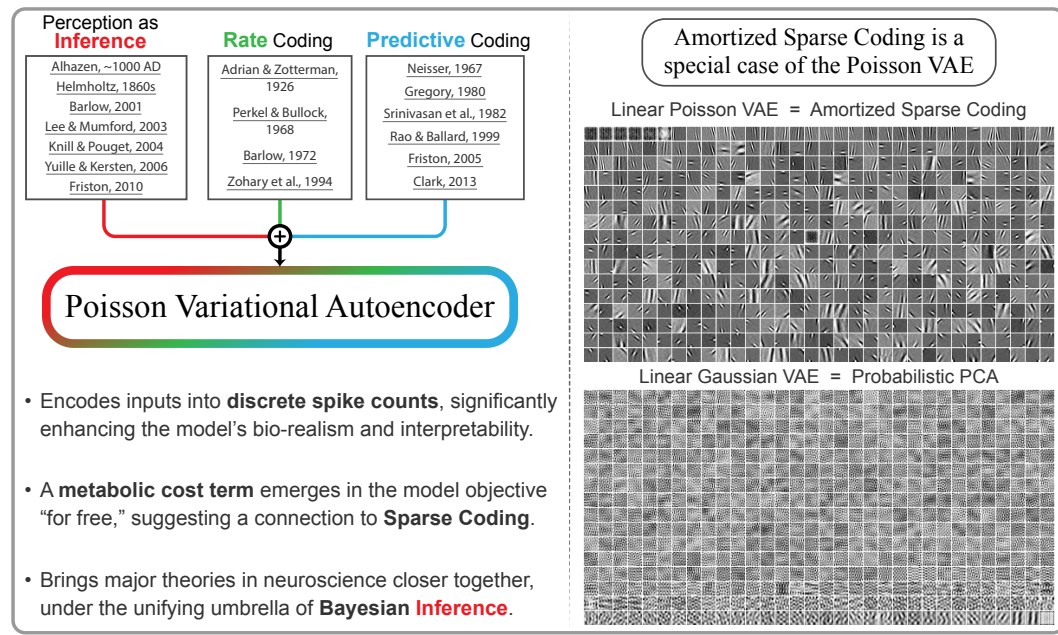

Figure 1: Graphical abstract. Introducing the Poisson Variational Autoencoder ($\mathcal{P}$-VAE), which draws on key concepts in neuroscience. When trained on natural image patches, $\mathcal{P}$-VAE with a linear decoder develops Gabor-like feature selectivity, reminiscent of Sparse Coding [3]. In sharp contrast, the standard Gaussian VAE learns the principal components [4]. Our code, data, and model checkpoints are available at this repository: https://github.com/hadivafaii/PoissonVAE

38th Conference on Neural Information Processing Systems (NeurIPS 2024).

# 1 Introduction

The study of artificial neural networks (ANN) and neuroscience has always been closely linked, driving advancements in both fields [5–10]. Despite the close proximity of the two fields, most ANN models deviate substantially from biological brains [11, 12]. A major challenge is designing models that not only perform well computationally but also exhibit "brain-like" structure and function. This is seen both as a goal for improving ANNs [13–15], and better understanding biological brains [8, 9, 16–19], which has recently been referred to as the *neuroconnectionist* research programme [20].

Drawing from neuroscience, a major guiding idea is that perception is a process of inference [21, 22], where the brain constructs a representation of the external world by inferring the causes of sensory inputs [23–26]. This concept is mirrored in "generative AI" where models learn the generative process underlying their inputs [27–29]. However, in this vein, there is a tension between small well-understood models that are directly inspired by cortex, such as sparse coding [3] and predictive coding [30], and deep generative models that perform well [31–34].

The variational autoencoder (VAE; [35, 36]) model family is a promising candidate for neuroconnectionist goals for multiple reasons. First, VAEs learn probabilistic generative models of their inputs and are grounded in Bayesian probability theory, providing a solid theoretical foundation that directly incorporates the concept of perceptual inference [10, 22]. Second, the VAE model family, specifically hierarchical VAEs, is broad with other generative models, such as diffusion models, understood as special cases of hierarchical VAEs [37–39]. Finally, VAEs learn representations that are similar to cortex [1, 2, 40], exhibit cortex-like topographic organization [41, 42], and make perceptual errors that mimic those of humans [43], indicating a significant degree of neural, organizational, and psychophysical alignment with the brain.

However, standard VAEs diverge from brains in the way they encode information. Biological neurons fire all-or-none action potentials [44], and are thought to represent information via firing rate [45–49]. These firing rates must be positive and generate discrete "spike" counts, which exhibit conditionally Poisson-like statistics in small counting windows [49–51]. In contrast, VAEs are typically parameterized with real-valued, continuous, Gaussian distributions [52].

**Contributions.** In this work, we address this discrepancy by introducing the Poisson Variational Autoencoder ($\mathcal{P}$-VAE), a novel architecture that combines perceptual inference with two other inspirations from neuroscience (Fig. 1). First, that information is encoded in the rates of discrete spike counts, which are approximately Poisson-distributed on short time intervals. And second, that feedforward connections encode deviations from expectations contained in feedback connections (Fig. 2a; [30, 53]). We introduce a reparameterization trick for Poisson samples (Algorithm 1), and derive the evidence lower bound (ELBO) objective for the $\mathcal{P}$-VAE (eq. (3)). Overall, we believe $\mathcal{P}$-VAE introduces a promising new model at the intersection of computational neuroscience and machine learning that offers several appealing features over existing VAE architectures:

- The $\mathcal{P}$-VAE loss derivation (eq. (3)) naturally results in a metabolic cost term that penalizes high firing rates, such that $\mathcal{P}$-VAE with a linear decoder implements amortized sparse coding (Fig. 2b). We validate this prediction empirically.
- $\mathcal{P}$-VAE largely avoids the prevalent posterior collapse issue, maintaining many more active latents compared to alternative VAE models (Table 1), especially the continuous ones.
- $\mathcal{P}$-VAE encodes its inputs in relatively higher dimensions, facilitating linear separability of categories in a downstream classification task with a much better ($5\times$) sample efficiency.

We evaluate these results on two natural image datasets and MNIST. The $\mathcal{P}$-VAE paves the way for the future development of interpretable hierarchical models that perform "brain-like" inference.

# 2 Background & Related work

**Perception as inference: connections to neuroscience and machine learning.** A centuries-old idea [21, 22], "perception as inference" argues that coherent perception of the world results from the unconscious inference over the causes of the senses. In other words, the brain learns a generative model of the sensory inputs. This has led to fruitful theoretical work in neuroscience [23, 54–56] and machine learning [57, 58], including VAEs [52]. See Marino [10] for a review.

**Efficient, predictive, and sparse coding.**    Another longstanding idea in neuroscience is that brains are adapted to the statistics of the environment. Efficient coding states that brains represent as much information about the environment as possible while minimizing neural resource use [59, 60].

Predictive coding [30, 61, 62] postulates that the brain generates a statistical prediction of its inputs, with feedforward networks carrying only the prediction errors or unexplained information [63]. More recently, ANNs based on predictive coding have been shown to capture a wide range of phenomena in biological neurons across the visual system [64, 65]. More broadly, prediction in time has emerged as an objective that lends itself to brain-like representations [66, 67].

Sparse coding (SC) is directly inspired by efficient coding, aiming to explain inputs as sparsely as possible [47, 68]. SC was the first unsupervised model to learn representations closely resembling the receptive fields of V1 neurons [3] and predicts an array of empirical features of neural activity [69–79]. SC is formalized with a generative model where neural activations $z$ are sampled from a sparsity-inducing prior, $z \sim p(z)$, and the input image $x$ is reconstructed as a linear combination of basis vectors $\mathbf{\Phi}$, plus additive Gaussian noise, $\hat{x} = \mathbf{\Phi}z + \varepsilon$. The SC loss is as follows:

$$\mathcal{L}_{\text{SparseCoding}}(x; \mathbf{\Phi}, z) = \|x - \mathbf{\Phi}z\|_2^2 + \beta \|z\|_1. \tag{1}$$

Commonly used algorithms for sparse coding include the locally competitive algorithm (LCA; [80]), which is a biologically plausible algorithm to optimize eq. (1), and iterative shrinkage-thresholding algorithm (ISTA; [81, 82]), which has shown robust performance in learning sparse codes given a fixed dictionary $\mathbf{\Phi}$.

**VAE objective.**    VAEs define a probabilistic generative model $p(x, z)$, where $x$ denotes the observed data and $z$ are some latent variables. The generative process samples $z$ from a prior distribution $p(z)$ and then generates the observed data $x$ from the conditional distribution $p_{\theta}(x|z)$, also known as the "decoder". The "encoder", $q_{\phi}(z|x)$, performs approximate inference on the inputs. Model parameters are learned by maximizing the evidence lower bound (ELBO) objective, which is derived from variational inference (see appendix B for the full set of derivations). The ELBO is given by:

$$\log p(x) \geq \mathbb{E}_{q_{\phi}(z|x)}\Big[\log p_{\theta}(x|z)\Big] - \mathcal{D}_{\text{KL}}\Big(q_{\phi}(z|x) \,\|\, p(z)\Big) = \mathcal{L}_{\text{VAE}}(x; \theta, \phi). \tag{2}$$

The first term captures the reconstruction performance of the decoder, and the second term, the "KL term," captures the divergence of the approximate posterior from the prior.

The specific form of these distributions is up to the practitioner. In standard VAEs, factorized Gaussians are typically used: $q = \mathcal{N}(z; \mu(x), \sigma^2(x))$ and $p = \mathcal{N}(z; 0, 1)$. The likelihood, $p_{\theta}(x|z)$, is also typically modeled as a Gaussian conditioned on a parameterized neural network $\text{dec}_{\theta}(z)$.

**Amortized inference in VAEs.**    A major contribution of VAEs is the idea of amortizing inference over the latents $z$ with a black box ANN [83, 84]. "Amortized" inference borrows a term from finance to capture the idea of spreading out costs—here, the cost of performing inference over multiple samples. In amortized inference, a neural network learns (during training) how to map a data sample to a distribution over latent variables given the sample. The cost is paid during training, but the trained model can then be used to perform inference on future samples efficiently. It has been argued that the brain performs amortized inference for computational efficiency [85].

**VAEs connection to biology.**    VAEs have been shown to contain individual latents that resemble neurons, capturing a wide range of the phenomena observed in visual cortical areas [40] and human perceptual judgments [43]. Like many other ANN models [86, 87], VAEs have been found to learn representations that are predictive of single-neuron activity in both the ventral [1] and dorsal [2] streams. However, unlike most ANNs, the mapping from certain VAEs to neural activity is incredibly sparse, even one-to-one in some cases [1, 2].

**Discrete VAEs.**    VAEs with discrete latent spaces, such as VQ-VAE [88] and Categorical VAE [89], are designed to capture complex data structures by mapping inputs to a finite set of latent variables. Unlike traditional VAEs that use continuous latent spaces, these models leverage discrete representations to enhance interpretability and can yield high performance with lower capacity [90].

---

**Algorithm 1** Reparameterized sampling (rsample) for Poisson distribution.

---

**Input:**
$\boldsymbol{\lambda} \in \mathbb{R}_{>0}^{B \times K}$         # rate parameter; $B$, batch size; $K$, latent dimensionality
$n\_\text{exp}$              # number of exponential samples to generate
temperature      # controls the sharpness of the thresholding

1: **procedure** RSAMPLE($\boldsymbol{\lambda}, n\_\text{exp}, \text{temperature}$)
2:      Exp $\leftarrow$ Exponential($\boldsymbol{\lambda}$)                 ▷ create exponential distribution
3:      $\Delta t \leftarrow$ Exp.rsample$((n\_\text{exp}, ))$      ▷ sample inter-event times, $\Delta t : [n\_\text{exp} \times B \times K]$
4:      times $\leftarrow$ **cumsum**($\Delta t, \text{dim=0}$)        ▷ compute arrival times, same shape as $\Delta t$
5:      indicator $\leftarrow$ sigmoid $\left( \frac{1-\text{times}}{\text{temperature}} \right)$      ▷ soft indicator for events within unit time
6:      $\boldsymbol{z} \leftarrow$ **sum**(indicator, dim=0)        ▷ event counts, or number of spikes, $\boldsymbol{z} : [B \times K]$
7:      **return** $\boldsymbol{z}$
8: **end procedure**

---

**VAEs connection to sparse coding.** Previous work has attempted to connect sparse coding and VAEs directly [91–93], with each approaching the problem differently. Geadah et al. [91] introduced sparsity-inducing priors (such as Laplace or Cauchy) and a linear decoder with an overcomplete latent space. Tonolini et al. [92] introduced a spike and slab prior into a modified ELBO, and Xiao et al. [93] added a sparse coding layer learned by ISTA to the latent space of a VQ-VAE. Notably, none of the three ended up minimizing the sparse coding loss. Two of the three maintain the linear generative model with an overcomplete latent space, but the ELBO in both requires an additional approximation step for the KL term [91, 92].

## 3    Introducing the Poisson Variational Autoencoder ($\mathcal{P}$-VAE)

Our main contribution is integrating Poisson-distributed latents into VAEs, where both the approximate posterior and the prior are parameterized as Poisson distributions. Critically, the latents $\boldsymbol{z}$ are no longer continuous variables, but rather they are discrete spike counts. To perform inference over discrete latents, we introduce a Poisson reparameterization trick. We then derive the KL term and obtain the full $\mathcal{P}$-VAE objective.

**Poisson reparameterization trick.** For a homogeneous Poisson process [94–96], given a window size $\Delta t = 1$, and rate $\lambda$, we can generate Poisson distributed counts by drawing randomly distributed wait-times from an exponential distribution with mean $1/\lambda$ and counting all events where the cumulative time is less than 1. Because the exponential distribution is trivially reparameterized [35], and PyTorch contains an implementation [97], we need only to approximate the hard threshold for comparing cumulative wait times with the window size. We accomplish this by replacing the indicator function with a sigmoid as in refs. [89, 98].

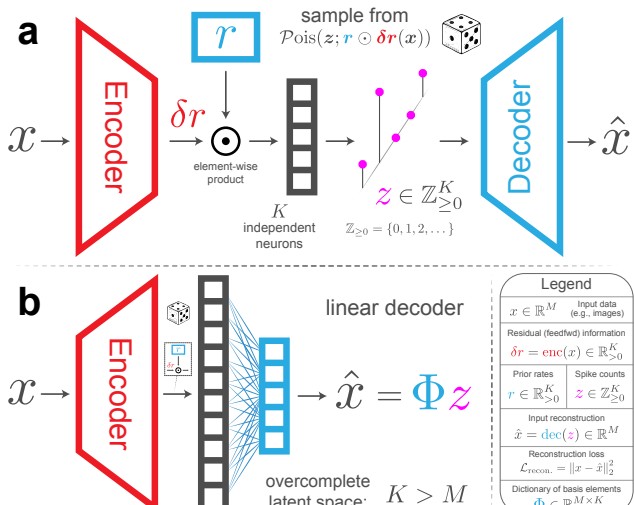

Figure 2: **(a)** Model architecture. Colored shapes indicate learnable model parameters, including the prior firing rates, $\boldsymbol{r}$. We color code the model's inference and generative components using red and blue, respectively. The $\mathcal{P}$-VAE encodes its inputs in discrete spike counts, $\boldsymbol{z}$, significantly enhancing its biological realism. **(b)** "Amortized Sparse Coding" is a special case within the $\mathcal{P}$-VAE model family: it's a $\mathcal{P}$-VAE with a linear decoder and an overcomplete latent space.

Algorithm 1 demonstrates the steps: Given a matrix of rates $\boldsymbol{\lambda}$, sample n_exp wait times $t_1, t_2, ...t_{n\_exp}$ for each element of $\boldsymbol{\lambda}$ by sampling from an exponential distribution with mean $1/\boldsymbol{\lambda}$. We then calculate the cumulative event times $S(n\_exp) = \sum_{j=1}^{n\_exp} t_j$, pass them through a sigmoid $\sigma(\frac{1-S}{\text{temperature}})$, and sum over samples to get event counts, $\boldsymbol{z}$. The temperature controls the sharpness of the thresholding. We adaptively scale the number of samples, n_exp, by keeping track of the maximum rate in each batch, $\lambda_{\max}$, and then use the inverse cumulative density function (cdf) for Poisson to find the number of samples, n_exp, such that $\text{cdf}(n\_exp; \lambda_{\max}) = 0.99999$.

At non-zero temperatures, our parameterization algorithm provides a continuous relaxation of the Poisson distribution. Figure 3 shows histograms of samples drawn using Algorithm 1 for rate $\lambda = 1$ and temperatures $T = 1.0, 0.1, 0.01$, and 0. The latter case ($T = 0$, true Poisson) is equivalent to `torch.poisson()`.

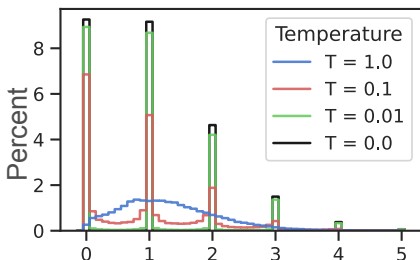

Figure 3: Relaxed Poisson distribution. Samples are drawn using Algorithm 1 for $\lambda = 1$. At non-zero temperatures, samples are non-integer, but approach the true Poisson distribution as $T \to 0$.

**$\mathcal{P}$-VAE architecture and residual parameterization.** The architecture of $\mathcal{P}$-VAE captures the interactions between feedforward and feedback connections that are present in all visual cortical areas [99, 100]. Feedforward areas carry sensory information and feedback connections are thought to carry modulatory signals such as attention [53] or prediction [30], which interact multiplicatively with feedforward inputs [53, 101].

$\mathcal{P}$-VAE embodies this idea by having the posterior rates depend on the prior, such that $\boldsymbol{r}_{\text{prior}} = \boldsymbol{r}$ and $\boldsymbol{r}_{\text{post.}} = \boldsymbol{r} \odot \boldsymbol{\delta r}(\boldsymbol{x})$, where $\odot$ is the Hadamard (element-wise) product. The prior rates, $\boldsymbol{r} \in \mathbb{R}^K$, are learnable parameters that capture expectations about the statistics of the input. The encoder outputs, $\boldsymbol{\delta r}(\boldsymbol{x}) \in \mathbb{R}^K$, capture *deviations* from the prior. Thus, $\mathcal{P}$-VAE models the interaction between prior expectations, and deviations from them, in a multiplicative and symmetric way. This results in a posterior, $q(\boldsymbol{z}|\boldsymbol{x}) = \mathcal{P}\text{ois}(\boldsymbol{z}; \boldsymbol{r} \odot \boldsymbol{\delta r}(\boldsymbol{x}))$, and prior, $p(\boldsymbol{z}) = \mathcal{P}\text{ois}(\boldsymbol{z}; \boldsymbol{r})$, where $\boldsymbol{z}$ is the spike count variable and $\mathcal{P}\text{ois}(z; \lambda) = \lambda^z e^{-\lambda}/z!$ is the Poisson distribution. Notably, this multiplicative relationship is maximally general, as any pair of positive variables, $\boldsymbol{r}_{\text{prior}}$, and $\boldsymbol{r}_{\text{post.}}$ can be expressed as a base variable, $\boldsymbol{r} := \boldsymbol{r}_{\text{prior}}$, multiplied by their relative ratio, $\boldsymbol{\delta r} := \boldsymbol{r}_{\text{post.}}/\boldsymbol{r}$. See Fig. 2a.

**$\mathcal{P}$-VAE loss function.** For a comprehensive derivation of the $\mathcal{P}$-VAE objective, see appendix B. Here, we report the final result:

$$\mathcal{L}_{\text{PVAE}} = \mathbb{E}_{\boldsymbol{z} \sim \mathcal{P}\text{ois}(\boldsymbol{z}; \boldsymbol{r} \odot \boldsymbol{\delta r})} \left[ \|\boldsymbol{x} - \text{dec}(\boldsymbol{z})\|_2^2 \right] + \sum_{i=1}^{K} r_i f(\delta r_i), \qquad (3)$$

where $\text{dec}(\cdot)$ is the decoder neural network, and $f(y) := 1 - y + y \log y$ (see supplementary Fig. 6).

**$\mathcal{P}$-VAE relationship to sparse coding.** The KL term in eq. (3) penalizes firing rates. Both $\boldsymbol{r}$ and $\boldsymbol{\delta r}$ are positive by definition, and $f(y) \geq 0$, strongly resembling the sparsity penalty in Olshausen and Field [3]. To make this connection more explicit, we make two additional assumptions (Fig. 2b):

1. The decoder is a linear generative model: $\hat{\boldsymbol{x}} = \boldsymbol{\Phi}\boldsymbol{z}$, with $\boldsymbol{x} \in \mathbb{R}^M$ and $\boldsymbol{\Phi} \in \mathbb{R}^{M \times K}$.
2. The latent space is overcomplete: $K > M$.

Because both $\mathbb{E}_{\boldsymbol{z} \sim \mathcal{P}\text{ois}(\boldsymbol{z}; \boldsymbol{\lambda})}[z_i]$ and $\mathbb{E}_{\boldsymbol{z} \sim \mathcal{P}\text{ois}(\boldsymbol{z}; \boldsymbol{\lambda})}[z_i z_j]$ have closed-form solutions (eq. (22)), the reconstruction term in eq. (3) can be computed analytically for a linear decoder, resulting in:

$$\mathcal{L}_{\text{SC-PVAE}}(\boldsymbol{x}; \boldsymbol{\delta r}, \boldsymbol{r}, \boldsymbol{\Phi}) = \|\boldsymbol{x} - \boldsymbol{\Phi}\boldsymbol{\lambda}\|_2^2 + \boldsymbol{\lambda}^T \text{diag}(\boldsymbol{\Phi}^T \boldsymbol{\Phi}) + \beta \sum_{i=1}^{K} r_i f(\delta r_i). \qquad (4)$$

where $\boldsymbol{\lambda} = \boldsymbol{r} \odot \boldsymbol{\delta r}(\boldsymbol{x})$ are the posterior firing rates, $f(y)$ is defined as above, and $\beta$ is a hyperparameter that scales the contribution of the KL term [102], and changes the sparsity penalty for the $\mathcal{P}$-VAE.

Table 1: Models considered in this paper.

| Discrete | | Continuous | |
|---|---|---|---|
| Poisson VAE ($\mathcal{P}$-VAE) | Categorical VAE ($\mathcal{C}$-VAE; [89, 98]) | Gaussian VAE ($\mathcal{G}$-VAE; [35, 36]) | Laplace VAE ($\mathcal{L}$-VAE; [40, 91]) |

The relationship between the linear $\mathcal{P}$-VAE loss (eq. (4)) and the sparse coding loss (eq. (1)) can now be seen. Both contain a term that minimizes the squared error of the reconstruction and a term (two terms for $\mathcal{P}$-VAE) that penalizes non-zero firing rates. Unlike prior work that directly implemented amortized sparse coding [91, 92], here the activity penalty naturally emerges from the derivations, and the only additional assumption was an overcomplete linear generative model. The inference is accomplished using a parameterized feed-forward neural network, $\boldsymbol{\delta r}(\boldsymbol{x})$, thus, it is amortized [83]. We call this specific case of $\mathcal{P}$-VAE "Amortized Sparse Coding" (Fig. 2b).

Note that a closed-form derivation of the reconstruction term is possible for any VAE with a linear decoder and a generating distribution that has a mean and variance (see eq. (21)).

This closed-form expression of the loss given a linear decoder is useful because we can see how different parameters contribute to the loss. Furthermore, we can compute gradients of the whole loss exactly, and use this to evaluate our Poisson reparameterization.

## 4 Experiments

To evaluate the $\mathcal{P}$-VAE, we perform three sets of experiments. First, we utilize the theoretical results for a linear decoder (eqs. (4) and (21)) to test the effectiveness of our reparameterization algorithm. We compare to alternative VAE models with established reparameterization tricks (e.g., Gaussian).

Second, to confirm $\mathcal{P}$-VAE with a linear decoder not only resembles amortized sparse coding but practically performs like sparse coding, we compare to standard and well-established sparse coding algorithms such as the locally competitive algorithm (LCA; [80]) and the widely-used iterative shrinkage-thresholding algorithm (ISTA; [81, 82]) to see if $\mathcal{P}$-VAE reproduces their results.

Third, we test the $\mathcal{P}$-VAE in a generic representation learning context and evaluate the geometry of learned representations for downstream tasks. For these experiments, both the encoder and decoder's architecture is a ResNet (see appendix C for full architecture and training details).

**Architecture notation.** We experimented with both convolutional and linear architectures. We highlight the encoder and decoder networks using red and blue, respectively. We use the $\langle \text{enc}|\text{dec}\rangle$ convention to clearly specify which architecture type was used. For example, $\langle \text{conv}|\text{lin}\rangle$ represents a model with a convolutional encoder and a linear decoder. Using this notation, we note that $\langle \text{lin}|\text{lin}\rangle$ and $\langle \text{conv}|\text{lin}\rangle$ architectures were used for the first and second sets of experiments, while $\langle \text{conv}|\text{conv}\rangle$ architectures were employed for the third.

**Alternative models.** We compare $\mathcal{P}$-VAE to both discrete and continuous VAEs (Table 1). Other than the traditional Gaussian, we compare to Laplace-distributed VAEs because previous work found the Laplace distribution supported robust sparse representations [40, 91]. Additionally, we compare to Categorical VAEs, trained using the Gumbel-Softmax trick [89, 98]. We use PyTorch's implementation which is based on Maddison et al. [98].

Finally, we test models where Gaussian latents are passed through an activation function before passing to the decoder. We call these models $\mathcal{G}$-VAE $_{+\text{act}}$, where $\text{act} \in \{\text{relu}, \text{exp}\}$, capturing other families of distributions (truncated Gaussian and log-normal). We include these to test the hypothesis that positive constraints (and not discrete latents) are the key contribution of Poisson [103].

**Datasets.** For sparse coding results, we use 101 natural images from the van Hateren dataset [104]. We tile the images to extract $16 \times 16$ patches and apply whitening and contrast normalization, as is typically done in sparse coding literature [3, 105]. To test the generalizability of our sparse coding results, we repeat these steps on CIFAR10 [106], a dataset we call CIFAR$_{16\times16}$. For the general representation learning results, we use MNIST. See appendix C for additional details.

Table 2: Reparameterized gradient estimators perform comparably to exact ones across datasets and encoder architectures (linear vs. convolutional). Exact gradients are only computable for linear decoders (see eqs. (21), (23) and (24)). Values represent percent drop in validation loss (lower is better), shown as mean±99% confidence interval calculated from $n = 5$ random initializations. The best-performing case was selected as the single best random seed for models of the same architecture and dataset across gradient methods (1 out of: 15 for $\mathcal{P}$-VAE, 10 for $\mathcal{G}$-VAE). See supplementary Fig. 7 for a visualization of the same data presented in this table. For actual loss values, see supplementary Table 5. EX: exact; MC: Monte Carlo; ST: straight-through [107].

| Model | | van Hateren | | CIFAR$_{16\times16}$ | | MNIST | |
|---|---|---|---|---|---|---|---|
| | | $\langle$lin$\|$lin$\rangle$ | $\langle$conv$\|$lin$\rangle$ | $\langle$lin$\|$lin$\rangle$ | $\langle$conv$\|$lin$\rangle$ | $\langle$lin$\|$lin$\rangle$ | $\langle$conv$\|$lin$\rangle$ |
| | EX | $0.6_{\pm.5}$ | $0.1_{\pm.1}$ | $0.0_{\pm.1}$ | $0.0_{\pm.0}$ | $0.1_{\pm.1}$ | $0.5_{\pm.6}$ |
| $\mathcal{P}$-VAE | MC | $0.0_{\pm.1}$ | $0.7_{\pm.1}$ | $0.2_{\pm.0}$ | $0.5_{\pm.1}$ | $0.7_{\pm.4}$ | $0.9_{\pm.5}$ |
| | ST | $7.3_{\pm.1}$ | $10.5_{\pm.1}$ | $9.1_{\pm.1}$ | $12.5_{\pm.1}$ | $8.1_{\pm.3}$ | $11.8_{\pm.2}$ |
| $\mathcal{G}$-VAE | EX | $0.1_{\pm.1}$ | $0.0_{\pm.0}$ | $0.0_{\pm.1}$ | $0.0_{\pm.0}$ | $0.1_{\pm.2}$ | $0.1_{\pm.2}$ |
| | MC | $0.1_{\pm.1}$ | $0.0_{\pm.0}$ | $0.1_{\pm.1}$ | $0.0_{\pm.0}$ | $0.4_{\pm.1}$ | $0.3_{\pm.1}$ |

**Statistical tests.** In the VAE literature, it is known that random seeds can have a large effect compared to architecture or regularization [108]. Therefore, we train each configuration using 5 different random initializations. We report 99% confidence intervals throughout, and perform paired $t$-tests, reporting significance for $p < 0.01$ (FDR corrected using the Benjamini-Hochberg method).

**Evaluating the Poisson reparameterization algorithm.** $\mathcal{P}$-VAE with a linear decoder has a closed form solution (eq. (4)), which lets us evaluate how well our reparameterized gradients perform compared to the exact ones. We compare our results to the gold-standard Gaussian (Table 2), as well as Categorical and Laplace VAEs (supplementary Table 5). In Table 2, we report the percent performance drop relative to the best fit, enabling meaningful comparisons across architectures and datasets. Monte Carlo sampling with Poisson reparameterization closely matches exact inference just like established methods for Gaussian and Laplace. In contrast, the straight-through (ST; [107]) estimator performs poorly (Table 2; see also supplementary Fig. 7).

**Annealing the temperature.** The temperature parameter ($T$) is a crucial hyperparameter in our Poisson reparameterization trick (Algorithm 1). To assess its impact, we followed standard practice [89] and annealed $T$ during the first half of training, starting from a large value ($T_{\text{start}} = 1$) and gradually decreasing it to a small value ($T_{\text{final}} = 0.05$ in the main paper). Figure 9 shows the performance on the van Hateren dataset as a function of various $T_{\text{final}}$, two architectures ($\langle$lin$|$lin$\rangle$ and $\langle$conv$|$lin$\rangle$), as well as two annealing schedules (linear vs. exponential; see inset). We find that final temperatures $T_{\text{final}} \leq 0.1$ and either annealing strategy work well.

During training, we maintain $T > 0$, which results in continuous (floating) latent variables, $\boldsymbol{z}$. At test time, we set $T = 0$ to produce genuine integer Poisson samples. Crucially, all reported results use $T = 0$ at test time. We also explored a "hard-forward" scheme during the latter half of training, where $T$ remains nonzero only in the backward pass. This *surrogate gradients* approach provides integer latents in the forward pass but, somewhat unexpectedly, underperformed our "relaxed Poisson" method (Fig. 9). These findings suggest that surrogate gradient methods might benefit from relaxing the hard-forward strategy during training. We believe this observation will be of particular interest to the spiking neural network community, which often relies on surrogate gradients for training.

**The $\mathcal{P}$-VAE learns basis vectors similar to those from sparse coding.** A major result from sparse coding is that it learns basis vectors (dictionaries) that resemble the "Gabor-like" receptive fields of cortical neurons [3, 109, 110]. Inspecting the dictionaries learned by different models demonstrates this is not trivial (Fig. 4). As expected from theoretical results [4], $\mathcal{G}$-VAE (top left) learn probabilistic PCA, but with many noisy elements. As demonstrated previously [40, 91], $\mathcal{L}$-VAE (lower left) learn Gabor-like elements. However, there are a large number of noisy basis vectors. It is of note that previous work did not show complete dictionaries for their results with Laplace latents [40, 91]. In contrast, $\mathcal{P}$-VAE (top middle) learns Gabor-like filters that cover space, orientation, and spatial

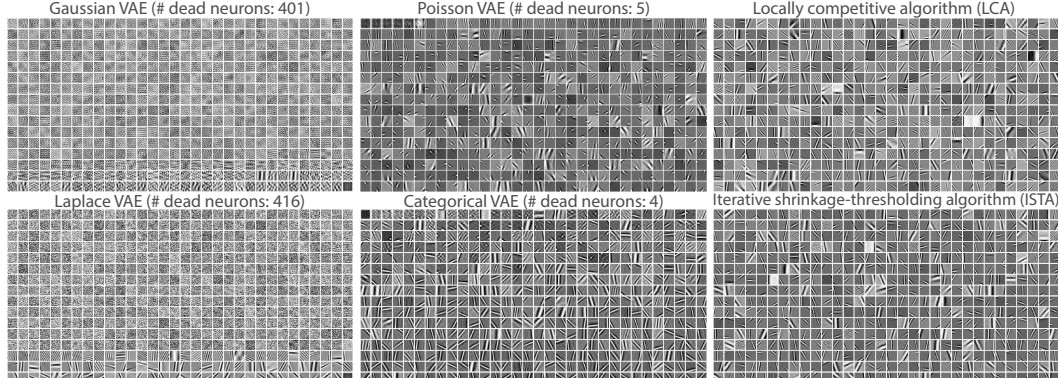

Figure 4: Learned basis elements for various $\langle \texttt{lin}|\texttt{lin}\rangle$ VAEs (first two columns) and standard sparse coding models (last column). There are a total of $K = 512$ elements, each made of $16 \times 16 = 256$ pixels (i.e., $\mathbf{\Phi} \in \mathbb{R}^{256 \times 512}$). Features are ordered from top-left to bottom-right, in ascending order of their associated $\texttt{KL}$ divergence ($\mathcal{P}$-VAE, $\mathcal{G}$-VAE, $\mathcal{L}$-VAE), or the magnitude of posterior $\texttt{logits}$ ($\mathcal{C}$-VAE). The sparse coding results (LCA and ISTA) are ordered randomly.

frequency. The quality is comparable to sparse coding dictionaries learned with LCA/ISTA (top/lower right panels). $\mathcal{C}$-VAE also learns Gabors, although there are significantly more noisy basis elements.

**The $\mathcal{P}$-VAE avoids posterior collapse.**    A striking feature of Fig. 4 is the sheer number of noisy basis vectors for both continuous VAEs ($\mathcal{G}$-VAE, $\mathcal{L}$-VAE). We suspected this reflected dead neurons with vanishing $\texttt{KL}$, which is indicative of a collapsed latent dimension that's no longer encoding information. To quantify this, we binned the distribution of $\texttt{KL}$ values and thresholded the resulting distribution at discontinuous points (see supplemental Fig. 10). Table 3 shows the results of this analysis for all VAEs with valid $\texttt{KL}$ terms. Across all datasets, both continuous VAEs suffered from large numbers of dead neurons, whereas $\mathcal{P}$-VAE largely avoided this problem. On both natural image datasets, $\mathcal{P}$-VAE had $\sim 2\%$ dead neurons compared to $\sim 80\%$ for $\mathcal{G}$-VAE and $\mathcal{L}$-VAE. Having a more expressive encoder slightly increases this percentage, but a dramatic difference between $\mathcal{P}$-VAE and continuous VAEs ($\mathcal{G}$-VAE, $\mathcal{L}$-VAE) persists.

**The $\mathcal{P}$-VAE learns sparse representations.**    To quantify whether $\mathcal{P}$-VAE learns sparse representations, we compared our VAE models to sparse coding trained with LCA and ISTA and quantified the lifetime sparsity [69]. The lifetime sparsity of the $j$-th latent is:

$$s_j = \left(1 - \frac{1}{N}\right)^{-1} \left(1 - \frac{1}{N} \frac{(\sum_i z_{ij})^2}{\sum_i z_{ij}^2}\right), \tag{5}$$

where $N$ is the number of images, and $z_{ij}$ is sampled from the posterior for the $i$-th image. Intuitively, $s_j = 1$ whenever neuron $j$ responds to a single stimulus out of the entire set (highly selective). In contrast, $s_j = 0$ whenever the neuron responds equally well to all stimuli indiscriminately.

Fig. 5a shows the reconstruction performance (MSE) compared to lifetime sparsity ($s$, eq. (5)) for all VAEs. Empty and solid circles represent $\langle \texttt{conv}|\texttt{lin}\rangle$ and $\langle \texttt{lin}|\texttt{lin}\rangle$ architectures, respectively. The $\mathcal{G}$-VAE finds good reconstructions (MSE = 71.49) but with low sparsity ($s = 0.37$). Because the $\mathcal{P}$-VAE $\texttt{KL}$ term explicitly penalizes rate (eq. (3)), we explored different $\beta$ values for $\mathcal{P}$-VAE with both $\langle \texttt{lin}|\texttt{lin}\rangle$ and $\langle \texttt{conv}|\texttt{lin}\rangle$ architectures (Fig. 5a, blue curves). This maps out rate-distortion curves, enabling us to compare the sparsity levels at which $\mathcal{P}$-VAE matches $\mathcal{G}$-VAE performance.

With a simpler (linear) encoder, $\langle \texttt{lin}|\texttt{lin}\rangle$ $\mathcal{P}$-VAE matches $\langle \texttt{conv}|\texttt{lin}\rangle$ $\mathcal{G}$-VAE performance while achieving $1.7\times$ greater sparsity at $\beta = 0.6$. A $\langle \texttt{conv}|\texttt{lin}\rangle$ $\mathcal{P}$-VAE further increases this gap to $2.4\times$ greater sparsity. Adding a $\texttt{relu}$ activation to $\mathcal{G}$-VAE also increases sparsity ($s = 0.69$). By comparing $\langle \texttt{lin}|\texttt{lin}\rangle$ and $\langle \texttt{conv}|\texttt{lin}\rangle$ $\mathcal{P}$-VAE models, we observe that enhancing encoder complexity for the same $\beta = 1$ (gray arrows) preserves MSE performance while achieving greater sparsity. This highlights how amortization quality can significantly influence rate-distortion curves [33, 111–113].

Table 3: Proportion of active neurons. All models considered in this table had a latent dimensionality of $K = 512$, with either $\langle$lin|lin$\rangle$ or $\langle$conv|lin$\rangle$ architectures. See also supplementary Fig. 10.

| Model | van Hateren | | CIFAR$_{16 \times 16}$ | | MNIST | |
|---|---|---|---|---|---|---|
| | linear | conv | linear | conv | linear | conv |
| $\mathcal{P}$-VAE | **0.984**±.011 | **0.819**±.041 | **0.999**±.002 | **0.928**±.045 | **0.537**±.008 | **0.426**±.011 |
| $\mathcal{L}$-VAE | 0.188±.000 | 0.222±.003 | 0.193±.003 | 0.230±.000 | 0.027±.000 | 0.034±.002 |
| $\mathcal{G}$-VAE | 0.218±.003 | 0.246±.000 | 0.105±.008 | 0.246±.000 | 0.027±.000 | 0.031±.000 |

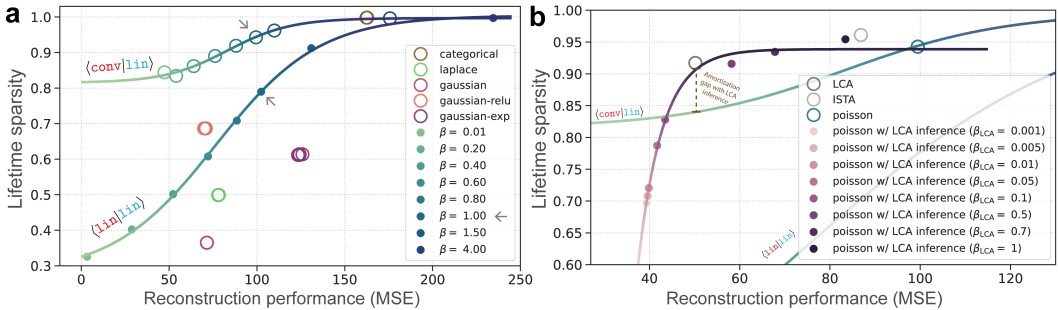

Figure 5: Reconstruction performance vs. sparsity of representations. **(a)** Results for the VAE model family. The curves are sigmoid fit to $\langle$lin|lin$\rangle$ and $\langle$conv|lin$\rangle$ $\mathcal{P}$-VAE results across varying $\beta$ values ($\beta$ from eq. (4)). Empty circles correspond to $\langle$conv|lin$\rangle$ architectures. **(b)** Amortization gap for $\mathcal{P}$-VAE (blue open circle) compared to sparse coding (LCA/ISTA). Solid points show results from applying the LCA inference algorithm to $\mathcal{P}$-VAE basis vectors at different sparsity levels ($\beta_{\text{LCA}}$ from eq. (1)). The purple curve is a sigmoid fit, and curves from part (a) are also included for comparison.

Does $\mathcal{P}$-VAE match the performance of traditional sparse coding trained with LCA or ISTA? Figure 5b compares $\mathcal{P}$-VAE to sparse coding models that were trained using a wide range of hyperparameters, and the best models were selected for each class (appendix C). $\mathcal{P}$-VAE achieves a similar sparsity to LCA and ISTA ($s = 0.94, 0.91$, and $0.96$, respectively), but the best LCA model drastically outperforms $\mathcal{P}$-VAE on MSE for similar levels of sparsity. This suggests our convolutional encoder is struggling to close the amortization gap. To test this hypothesis, we performed LCA inference on basis elements learned by $\mathcal{P}$-VAE (Fig. 5b curve/solid points). We explored a range of hyperparameters to determine whether the MSE improved for similar sparsity levels. Indeed, LCA inference using $\mathcal{P}$-VAE dictionary was able to nearly match the performance of sparse coding LCA for similar levels of sparsity. This confirms our hypothesis that a large amortization gap remains for the specific encoder architectures we tested, highlighting the need for improved inference algorithms/architectures [112].

**The $\mathcal{P}$-VAE is more sample efficient in downstream tasks.** To assess downstream performance, we trained $\langle$conv|conv$\rangle$ VAE models with a $K = 10$ latent dimension on MNIST (see supplementary Fig. 12 for generated samples and reconstructions from these models). We then extracted representations from the trained encoders and evaluated their ability to classify MNIST digits. We define representations as mean vectors $\boldsymbol{\mu}$ for continuous VAEs ($\mathcal{G}$-VAE, $\mathcal{L}$-VAE) following conventions in the VAE literature [108], and use $\log \boldsymbol{\delta r}$ for $\mathcal{P}$-VAE, and logits for $\mathcal{C}$-VAE.

We split the MNIST validation set into two 5,000 sample sets, used as train/test sets for this task. We train K-nearest neighbors (KNN) classifiers with a varying number of limited supervised samples ($N = 200, 1000, 5000$) drawn without replacement from the first set (train), to measure classification accuracy on the withheld set (test). KNN is nonparametric, and its performance is directly influenced by the geometry of representations by explicitly capturing the distance between encoded samples [114]. We find that using only $N = 200$ samples, $\mathcal{P}$-VAE achieves $\sim 82\%$ accuracy in held out data; whereas, $\mathcal{G}$-VAE achieves the same level of accuracy at $N = 1000$ samples (Table 4). By this measure, $\mathcal{P}$-VAE is $5\times$ more sample efficient. But from Alleman et al. [115], we know that the choice of activation function changes the geometry of learned representations. Therefore, we also tested $\mathcal{G}$-VAE models with an activation function (relu and exp) applied to latents after sampling from the

Table 4: Geometry of representations ($K = 10$ only; see Table 6 for the full set of results).

| Latent dim. | Model | KNN classification ($N$, # labeled samples) | | | Shattering dim. |
| | | $N = 200$ | $N = 1,000$ | $N = 5,000$ | |
| --- | --- | --- | --- | --- | --- |
| $K = 10$ | $\mathcal{P}$-VAE | $\mathbf{0.815}_{\pm.002}$ | $\mathbf{0.919}_{\pm.001}$ | $\mathbf{0.946}_{\pm.017}$ | $\mathbf{0.797}_{\pm.009}$ |
| | $\mathcal{C}$-VAE | $0.705_{\pm.002}$ | $0.800_{\pm.002}$ | $0.853_{\pm.040}$ | $\mathbf{0.795}_{\pm.006}$ |
| | $\mathcal{L}$-VAE | $0.757_{\pm.003}$ | $0.869_{\pm.002}$ | $\mathbf{0.924}_{\pm.028}$ | $0.751_{\pm.008}$ |
| | $\mathcal{G}$-VAE | $0.673_{\pm.003}$ | $0.813_{\pm.002}$ | $0.891_{\pm.033}$ | $0.758_{\pm.007}$ |
| | $\mathcal{G}$-VAE $_{+\text{relu}}$ | $0.694_{\pm.003}$ | $0.817_{\pm.003}$ | $0.877_{\pm.045}$ | $0.762_{\pm.007}$ |
| | $\mathcal{G}$-VAE $_{+\text{exp}}$ | $0.642_{\pm.003}$ | $0.784_{\pm.002}$ | $0.863_{\pm.032}$ | $0.737_{\pm.008}$ |

posterior. This biological constraint improved $\mathcal{G}$-VAE, but it still underperformed $\mathcal{P}$-VAE (Table 4). We also found this result held for higher dimensional latent spaces (supplementary Table 6).

In supplementary analyses (Fig. 11), we evaluated the representations using logistic regression trained on the full dataset. For larger latent dimensionalities ($K = 50, 100$), $\mathcal{P}$-VAE outperformed all other VAEs, but at lower dimensionalities ($K = 10$), it underperforms both $\mathcal{G}$-VAE and $\mathcal{L}$-VAE.

**The $\mathcal{P}$-VAE learns representations with higher dimensional geometry.** The preceding results are indicative of substantial differences in the geometry of the representations learned by $\mathcal{P}$-VAE compared to other VAE families (Table 4). To test this more explicitly, we calculated the "shattering dimensionality" of the latent space [116–118]. Shattering dim measures the average accuracy over all possible pairwise classification tasks. This is called "shattering" because if the model shatters data points around into a high dimensional space, they will become more linearly separable. For MNIST with 10 classes, there are $\binom{10}{5} = 252$ possible classifications. We trained logistic regression on the entire training set to classify each of the 252 arbitrary splits and measured the average performance on the entire validation set. The far right column of Table 4 shows the measured shattering dims. For $K = 10$, the shattering dim was significantly higher for discrete VAEs ($\mathcal{P}$-VAE, $\mathcal{C}$-VAE). For higher dimensional latent spaces $\mathcal{P}$-VAE strongly outperformed alternative models (Table 6).

## 5  Conclusions

In this paper, we introduced the $\mathcal{P}$-VAE, a generative model that encodes inputs into discrete spike counts and unifies established theoretical concepts in neuroscience with modern machine learning. We introduced a Poisson reparameterization algorithm and derived the ELBO for Poisson-distributed latent variables. The $\mathcal{P}$-VAE objective results in a KL term that penalizes firing rates, like sparse coding. We showed that $\mathcal{P}$-VAE with a linear decoder reduces to amortized sparse coding. We evaluated the representations on downstream classification tasks and found that $\mathcal{P}$-VAE encodes its inputs in a higher dimensional space, enabling good linear separability between classes.

**Limitations.** $\mathcal{P}$-VAE samples Poisson latents. Although this is inspired by the statistics of spike counts in the brain over short time intervals [50], there are deviations from Poisson throughout the cortex over longer time windows [51]. We discuss this point in appendix A. A second limitation is the amortization gap between our current implementation of $\mathcal{P}$-VAE and traditional sparse coding. This could likely be closed with more expressive encoders [119] or through iterative inference [113, 120], but it is an open area of research [112].

**Neuroscience implications and future directions.** Like biological neurons, the $\mathcal{P}$-VAE generates spikes. This non-negative, discrete representational form closely parallels neuronal spiking activity. Therefore, the $\mathcal{P}$-VAE can be more directly compared to neuronal circuits than unconstrained, continuous VAEs. This analogy facilitates in silico perturbation experiments (e.g., "stimulating" or "silencing" $\mathcal{P}$-VAE neurons) to mirror in vivo causal manipulations. It also allows applying methods like *Most Exciting Inputs* (MEI; [121]), which assume non-negative activations. Future work could explore hierarchical $\mathcal{P}$-VAEs, finding a sweet spot between interpretability and performance. Overall, the biologically inspired representational form of $\mathcal{P}$-VAE brings computational modeling closer to experimental neuroscience and opens new avenues for advancing NeuroAI research [13, 20].

## 6 Code and data

Our code, data, and model checkpoints are available here: https://github.com/hadivafaii/PoissonVAE.

## 7 Acknowledgments

This work was supported by the National Institute of Health under award number NEI EY032179. Additionally, this material is based upon work supported by the National Science Foundation Graduate Research Fellowship Program under Grant No. DGE-1752814 (DG). Any opinions, findings, conclusions, or recommendations expressed in this material are those of the author(s) and do not necessarily reflect the views of the National Science Foundation. We thank our anonymous reviewers for their helpful comments, and the developers of the software packages used in this project, including PyTorch [97], NumPy [122], SciPy [123], scikit-learn [124], pandas [125], matplotlib [126], and seaborn [127].

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

# A    Are real neurons truly Poisson?

In this section, we discuss empirical and theoretical observations from neuroscience that motivated our Poisson assumption.

"Poisson-like" noise in neuroscience has a long history. It begins with observations that neurons do not fire the same sequence of spikes to repeated presentations of the same input, and that the variance is proportional to the mean [128, 129], and was followed by the observation that for short counting windows, that proportionality is one [49, 50, 130–132]. Larger windows and higher visual areas are notably super-Poisson, but that can be attributed to a modulation of the rate of an inhomogeneous Poisson process [51].

In other words, neurons are conditionally Poisson, not marginally Poisson [133].

Spike-generation, it is argued, is not noisy [134–136], but synaptic noise [137], or noise on the membrane potential, can create a Poisson-like distribution of spikes [138]. An important caveat is that the well-known example of precision in spike generation by Mainen and Sejnowski [134] is effectively captured by a Poisson-process Generalized Linear Model (GLM; Weber and Pillow [139]). However, this precision relies on a Bernoulli approximation to a Poisson process, allowing only 0 or 1 spikes. There is a widely-held misconception that precise timing cannot be produced by spike-rate models, but inhomogeneous rate models can produce precise spiking patterns at high time resolution [140]. In contrast, recent work has shown that correlations in excitatory inputs drive Poisson-like variability, explaining the widespread observation of Poisson-like noise in real neurons [141].

In summary, neurons are not literally Poisson, but it is a good choice. To set up the ELBO, one has to choose an approximate posterior and prior. Because spike counts are integer and cannot be negative, Poisson is a more natural choice than Gaussian without knowing anything about neural firing statistics. Here, we found that the Poisson assumption led to a model with interesting theoretical and empirical properties, where sparse coding emerged from the ELBO with Poisson.

Extending the $\mathcal{P}$-VAE to hierarchical architectures [2, 33, 34, 142] will make the latents conditionally Poisson, but not marginally Poisson (as they are modulated by top-down rates). Further extensions could implement doubly-stochastic spike generation [51, 143].

# B    Full derivations

In this section, we provide a self-contained and pedagogical introduction to VAEs, derive the $\mathcal{P}$-VAE loss function, and highlight how combining Poisson-distributed latents with predictive coding leads to the emergence of a metabolic cost term in the $\mathcal{P}$-VAE loss. For the case of a linear decoder, the reconstruction loss assumes a closed-form solution. This means we can compute the gradients analytically, which we can then use to evaluate the Poisson reparameterization trick.

## B.1    Deriving the evidence lower bound (ELBO) loss

For completeness, let's first go over the basics. This section will provide a quick refresher on variational inference and how to derive the VAE loss from scratch. Assume the data $\boldsymbol{x} \in \mathbb{R}^M$ and $K$-dimensional latent variables $\boldsymbol{z}$ are jointly distributed as $p(\boldsymbol{x}, \boldsymbol{z})$, with the data generated through the following process:

$$p(\boldsymbol{x}) = \int p(\boldsymbol{x}, \boldsymbol{z}) \, d\boldsymbol{z} = \int p(\boldsymbol{x}|\boldsymbol{z})p(\boldsymbol{z}) \, d\boldsymbol{z}, \tag{6}$$

In Bayesian posterior inference, the goal is to identify which latents $\boldsymbol{z}$ are likely given data $\boldsymbol{x}$. In other words, we want to approximate $P(\boldsymbol{z}|\boldsymbol{x})$, the optimal but (typically) intractable posterior distribution.

### B.1.1    Variational inference and VAE loss function

To achieve approximate Bayesian inference, a common approach is to define a family of variational densities $\mathcal{Q}$ and find a member $q(\boldsymbol{z}|\boldsymbol{x}) \in \mathcal{Q}$ such that it sufficiently approximates the optimal posterior [144]. We call $q(\boldsymbol{z}|\boldsymbol{x})$ the *approximate posterior*. The general aim of variational inference (VI) can be summarized as follows:

$$\text{VI}: \quad \text{find a } q(\boldsymbol{z}|\boldsymbol{x}) \in \mathcal{Q} \text{ such that } q(\boldsymbol{z}|\boldsymbol{x}) \text{ is a good approximation of } p(\boldsymbol{z}|\boldsymbol{x}). \tag{7}$$

The goodness of our approximate posterior, or its closeness to the true posterior, is measured using the Kullback-Leibler (KL) divergence:

$$q^* = \underset{q \in \mathcal{Q}}{\operatorname{argmin}} \, \mathcal{D}_{\text{KL}}\Big( q(\boldsymbol{z}|\boldsymbol{x}) \, \| \, p(\boldsymbol{z}|\boldsymbol{x}) \Big). \tag{8}$$

We cannot directly optimize eq. (8), because $p(\boldsymbol{z}|\boldsymbol{x})$ is often intractable. Instead, we rearrange some terms and arrive at the following loss function:

$$\mathcal{L}_{\text{NELBO}}(q) = -\mathbb{E}_{\boldsymbol{z} \sim q(\boldsymbol{z}|\boldsymbol{x})}\Big[ \log p(\boldsymbol{x}|\boldsymbol{z}) \Big] + \mathcal{D}_{\text{KL}}\Big( q(\boldsymbol{z}|\boldsymbol{x}) \, \| \, p(\boldsymbol{z}) \Big). \tag{9}$$

NELBO stands for negative ELBO, also known as "variational free energy." Notably, finding a $q \in \mathcal{Q}$ that minimizes $\mathcal{L}_{\text{NELBO}}(q)$ in eq. (9) is equivalent to finding the optimal $q^*$ in eq. (8).

The first term in eq. (9), often called the reconstruction term, captures the likelihood of the observed data $\boldsymbol{x}$, given latents $\boldsymbol{z}$, under the approximate posterior. For all our VAE models, we assume a Gaussian conditional likelihood with a fixed variance, as is typically done in the literature. This approximates the reconstruction term as the mean squared error between input data and their reconstructed version. The second term, known as the KL term, is more interesting. This term can assume very different forms depending on the distribution used.

## B.2  The KL term

In this section, we will derive closed-form expressions for the KL term for different choices of the distributions $q(\boldsymbol{z}|\boldsymbol{x})$ and $p(\boldsymbol{z})$. Specifically, we will focus on Gaussian and Poisson parameterizations.

**Predictive coding assumption.**  We will draw inspiration from predictive coding and assume that the bottom-up inference pathway only encodes the residual information relative to the top-down, or predicted information. We will apply this idea to both Gaussian and Poisson cases, and find that only in the Poisson case, the outcome becomes interpretable and resembles sparse coding objective.

### B.2.1  KL term derivation: Gaussian

Let $q(\boldsymbol{z}|\boldsymbol{x}) = \mathcal{N}(\boldsymbol{z}; \boldsymbol{\mu}_q(\boldsymbol{x}), \boldsymbol{\sigma}_q(\boldsymbol{x}))$ and $p(\boldsymbol{z}) = \mathcal{N}(\boldsymbol{z}; \boldsymbol{\mu}_p, \boldsymbol{\sigma}_p)$, where the mean and variance are either outputs of the encoder network or parameters of the decoder network.

Now, let us implement the predictive coding assumption, where the encoder only keeps track of residual information that is not already contained in the prior information. Mathematically, this idea can be formalized as follows:

$$\begin{aligned} \boldsymbol{\mu}_p &\to \boldsymbol{\mu}, & \boldsymbol{\mu}_q &\to \boldsymbol{\mu} + \boldsymbol{\delta\mu} \\ \boldsymbol{\sigma}_p &\to \boldsymbol{\sigma}, & \boldsymbol{\sigma}_q &\to \boldsymbol{\sigma} \cdot \boldsymbol{\delta\sigma} \end{aligned} \tag{10}$$

With these modifications, the Gaussians KL term becomes:

$$\mathcal{D}_{\text{KL}}\left( q \, \| \, p \right) = \frac{1}{2}\Big( \frac{\boldsymbol{\delta\mu}^2}{\boldsymbol{\sigma}^2} + \boldsymbol{\delta\sigma}^2 - \log \boldsymbol{\delta\sigma}^2 - \boldsymbol{1} \Big). \tag{11}$$

In standard Gaussian VAEs, the prior has no learnable parameter. Instead, we have $\boldsymbol{\mu} \to \boldsymbol{0}$ and $\boldsymbol{\sigma} \to \boldsymbol{1}$. Therefore, the final form of the KL term for a standard Gaussian VAE is:

$$\mathcal{D}_{\text{KL}}\left( q \, \| \, \mathcal{N}(\boldsymbol{0}, \boldsymbol{1}) \right) = \frac{1}{2}\Big( \boldsymbol{\delta\mu}^2 + \boldsymbol{\delta\sigma}^2 - \log \boldsymbol{\delta\sigma}^2 - \boldsymbol{1} \Big). \tag{12}$$

We observe that the KL term vanishes when $\boldsymbol{\delta\mu} \to \boldsymbol{0}$ and $\boldsymbol{\delta\sigma} \to \boldsymbol{1}$. This happens whenever no new information is propagated through the encoder, a phenomenon known as posterior collapse.

Other than this trivial observation, eq. (12) does not really lend itself to interpretation. In contrast, will show below that a Poisson parameterization of VAEs leads to a much more interpretable outcome for the KL term.

### B.2.2    KL term derivation: Poisson

Now suppose $q(z|\boldsymbol{x}) = \mathcal{P}\mathrm{ois}(z; r\delta r(\boldsymbol{x}))$, and $p(z) = \mathcal{P}\mathrm{ois}(z; r)$, where $z$ is literally the spike count of a single latent dimension—or shall we say, neuron?

In the Poisson case, the KL term becomes more interpretable, as we will show below. Recall that the Poisson distribution for a single variable $z$, given rate $\lambda \in \mathbb{R}_{>0}$, is given by:

$$\mathcal{P}\mathrm{ois}(z; \lambda) = \frac{\lambda^z e^{-\lambda}}{z!}. \tag{13}$$

Plug this expressions into the KL divergence definition to get:

$$
\begin{aligned}
\mathcal{D}_{\mathrm{KL}}\left(q \,\|\, p\right) &= \mathbb{E}_{z\sim q}\left[\log \frac{q}{p}\right] \\
&= \mathbb{E}_{z\sim q}\left[\log \frac{(r\delta r)^z e^{-r\delta r}/z!}{r^z e^{-r}/z!}\right] \\
&= \mathbb{E}_{z\sim q}\left[\log \left(\left(\frac{r\delta r}{r}\right)^z e^{-r\delta r + r}\right)\right] \\
&= \mathbb{E}_{z\sim q}\left[\log \delta r^z + \log e^{-r\delta r + r}\right] \\
&= \mathbb{E}_{z\sim q}\left[z \log \delta r - r\delta r + r\right] \\
&= \mathbb{E}_{z\sim q}\left[z\right] \log \delta r - r\delta r + r \\
&= r\delta r \log \delta r - r\delta r + r \\
&= r\left(1 - \delta r + \delta r \log \delta r\right) \\
&= r f(\delta r),
\end{aligned}
\tag{14}
$$

where we have define $f(y) := 1 - y + y \log y$.

To examine the behavior of the Poisson KL term, we assume $\delta r = 1 + \epsilon$, where $\epsilon \ll 1$, then Taylor expand $f$. Calculating the first and second derivatives of $f(y) = 1 - y + y \log y$ gives $f'(y) = \log y$ and $f''(y) = 1/y$. Thus:

$$
\begin{aligned}
f(1 + \epsilon) &= f(1) + \epsilon f'(1) + \frac{\epsilon^2}{2!} f''(1) + \mathcal{O}(\epsilon^3) \\
&= 0 + 0 + \frac{\epsilon^2}{2!} + \mathcal{O}(\epsilon^3) \\
&\approx \frac{1}{2}\epsilon^2
\end{aligned}
\tag{15}
$$

Plug this back into eq. (14) to get:

$$
\begin{aligned}
\mathcal{D}_{\mathrm{KL}}\left(q \,\|\, p\right) &= r f(\delta r) \\
&= r f(1 + \epsilon) \\
&\approx \frac{1}{2} r \epsilon^2.
\end{aligned}
\tag{16}
$$

For small deviations $\epsilon$, the KL term simplifies to the product of the prior firing rate, $r$, and $\epsilon^2$. See Fig. 6 for a visualization of the full function, $f(\delta r) = 1 - \delta r + \delta r \log \delta r$, along with its quadratic approximation near $\delta r = 1$.

In general, there are two ways to minimize the KL term: dead prior neurons ($r \to 0$), or posterior collapse ($\delta r \to 1$).

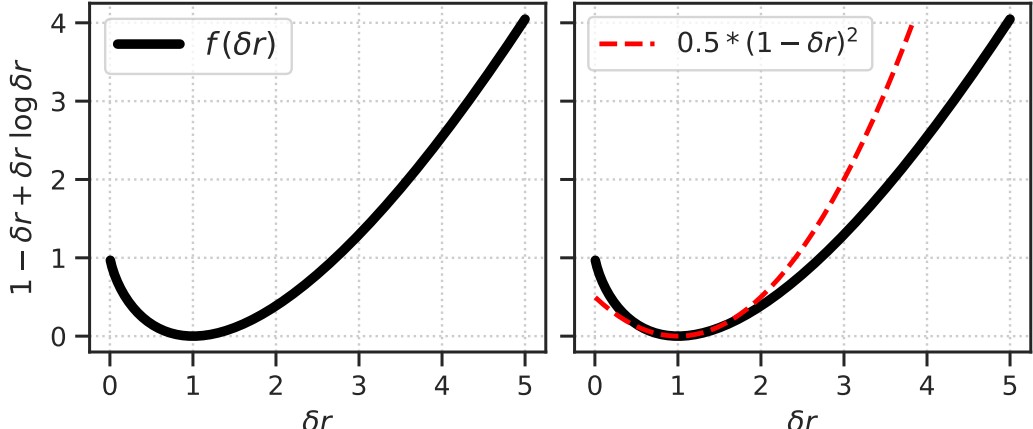

Figure 6: Left, residual term $f(\delta r)$ from eq. (14). Right, quadratic approximation of $f$ from eq. (15).

Together with the reconstruction loss, the NELBO for a 1-dimensional $\mathcal{P}$-VAE reads:

$$\mathcal{L}_{\text{PVAE}}\left(r, \delta r\right) = \mathcal{L}_{\text{recon.}}\left(r, \delta r\right) + r\left(1 - \delta r + \delta r \log \delta r\right). \tag{17}$$

Finally, it is easy to show that for $K$-dimensional latent space, eq. (14) generalizes to:

$$\mathcal{D}_{\text{KL}}\Big(\mathcal{P}\text{ois}(\boldsymbol{z}; \boldsymbol{r} \odot \boldsymbol{\delta r}(\boldsymbol{x})) \,\big\|\, \mathcal{P}\text{ois}(\boldsymbol{z}; \boldsymbol{r})\Big) = \boldsymbol{r} \cdot f(\boldsymbol{\delta r}), \tag{18}$$

where $\odot$ and $\cdot$ denote the Hadamard (element-wise) and vector products, respectively.

### B.3 Connection to sparse coding

Equation (17) mirrors sparse coding due to the presence of the firing rate in the objective function. Furthermore, it follows the principle of predictive coding by design. Thus, our Poisson formulation of VAEs effectively unifies these two major themes in theoretical neuroscience. Let's explore this curious connection to sparse coding more closely below.

### B.4 Statistically independent neurons

Suppose our $\mathcal{P}$-VAE has $K$ statistically independent neurons, and $\boldsymbol{z} \in \mathbb{Z}_{\geq 0}^{K}$ is the spike count variable, where $\mathbb{Z}_{\geq 0} = \{0, 1, 2, \dots\}$ is the set of non-negative integers. Let us use bold font $\boldsymbol{r}$ and $\boldsymbol{\delta r}$ to refer to the firing rate vectors of the representation and error units, respectively. Recall that we allowed these variables to interact in a multiplicative way to construct the posterior rates, $\lambda_i(\boldsymbol{x}) = r_i \delta r_i(\boldsymbol{x})$. More explicitly, we have:

$$q(\boldsymbol{z}|\boldsymbol{x}) = \mathcal{P}\text{ois}(\boldsymbol{z}; \boldsymbol{r} \odot \boldsymbol{\delta r}) = \prod_{i=1}^{K} \mathcal{P}\text{ois}(z_i; r_i \delta r_i) = \prod_{i=1}^{K} \frac{\lambda_i^{z_i} e^{-\lambda_i}}{z_i!},$$

$$p(\boldsymbol{z}) = \mathcal{P}\text{ois}(\boldsymbol{z}; \boldsymbol{r}) = \prod_{i=1}^{K} \mathcal{P}\text{ois}(z_i; r_i) = \prod_{i=1}^{K} \frac{r_i^{z_i} e^{-r_i}}{z_i!}. \tag{19}$$

Note that, unlike a standard Gaussian VAE, the prior in $\mathcal{P}$-VAE is parameterized using $\boldsymbol{r}$, which is learned from data along with the other parameters. Similar to standard Gaussian VAEs, $\boldsymbol{\delta r}(\boldsymbol{x})$ is parameterized as a neural network.

## B.5 Linear decoder

Following the sparse coding literature [3], we will now assume our decoder generates the input image $\boldsymbol{x} \in \mathbb{R}^M$ as a linear sum of $K$ basis elements, $\boldsymbol{\Phi} \in \mathbb{R}^{M \times K}$. Additionally, we choose a diagonal Gaussian distribution with fixed variance as our conditional likelihood, resulting in a mean squared error between the input $\boldsymbol{x}$, and its reconstruction $\boldsymbol{\Phi z}$.

Given these assumptions, the reconstruction loss for a VAE with approximate posterior $q$ can be expressed as follows:

$$\mathcal{L}_{\text{recon.}}(\boldsymbol{x}; q) = \mathbb{E}_{\boldsymbol{z} \sim q(Z|X=\boldsymbol{x})} \left[ \|\boldsymbol{x} - \boldsymbol{\Phi z}\|_2^2 \right]. \tag{20}$$

For a linear decoder, the reconstruction term $\|\boldsymbol{x} - \boldsymbol{\Phi z}\|_2^2$ contains only the first and second moments of $\boldsymbol{z}$. Consequently, the expectation in eq. (20) can be analytically computed. This results in a close-form expression for the reconstruction loss, and consequently, its gradients as well.

In general, whenever the VAE decoder is linear, the following result holds:

$$\boxed{\mathcal{L}_{\text{recon.}}(\boldsymbol{x}; q, \boldsymbol{\Phi}) = \|\boldsymbol{x} - \boldsymbol{\Phi} \mathbb{E}_q[Z]\|_2^2 + \text{Var}_q[Z]^T \text{diag}(\boldsymbol{\Phi}^T \boldsymbol{\Phi}).} \tag{21}$$

Note that a linear decoder is the only assumption we needed to obtain this closed-form solution. There are no restrictions on the form of the encoder: it can be linear, or as complicated as we want. We only have to compute the mean and variance of the posterior.

Specifically, for the Poisson case, we only need to know the following expectation values:

$$\begin{aligned} \mathbb{E}_{\boldsymbol{z} \sim \mathcal{P}\text{ois}(\boldsymbol{z}; \boldsymbol{\lambda})} \left[ z_i \right] &= \lambda_i, \\ \mathbb{E}_{\boldsymbol{z} \sim \mathcal{P}\text{ois}(\boldsymbol{z}; \boldsymbol{\lambda})} \left[ z_i z_j \right] &= \lambda_i \lambda_j + \delta_{ij} \lambda_i. \end{aligned} \tag{22}$$

Here are the reconstruction losses for both Poisson and Gaussian VAEs with linear decoders, put side-by-side for comparison:

$$\boxed{\begin{aligned} \text{Poisson:} \qquad & \mathcal{L}_{\text{recon.}}(\boldsymbol{x}; \boldsymbol{\lambda}, \boldsymbol{\Phi}) = \|\boldsymbol{x} - \boldsymbol{\Phi \lambda}\|_2^2 + \boldsymbol{\lambda}^T \text{diag}(\boldsymbol{\Phi}^T \boldsymbol{\Phi}), \\ \text{Gaussian:} \qquad & \mathcal{L}_{\text{recon.}}(\boldsymbol{x}; \boldsymbol{\mu}, \boldsymbol{\sigma}, \boldsymbol{\Phi}) = \|\boldsymbol{x} - \boldsymbol{\Phi \mu}\|_2^2 + (\boldsymbol{\sigma}^2)^T \text{diag}(\boldsymbol{\Phi}^T \boldsymbol{\Phi}). \end{aligned}} \tag{23}$$

Given these assumptions, the NELBO (eq. (9)) for $\mathcal{P}$-VAE with a linear decoder becomes:

$$\boxed{\mathcal{L}_{\text{SC-PVAE}}(\boldsymbol{x}; \boldsymbol{\delta r}, r, \boldsymbol{\Phi}) = \|\boldsymbol{x} - \boldsymbol{\Phi \lambda}\|_2^2 + \boldsymbol{\lambda}^T \text{diag}(\boldsymbol{\Phi}^T \boldsymbol{\Phi}) + \beta \sum_{i=1}^{K} r_i f(\delta r_i).} \tag{24}$$

Recall that we have $f(y) = 1 - y + y \log y$ (see Fig. 6). We introduced the $\beta$ term here to control the trade-off between the reconstruction and the KL term [102]. Additionally, we dropped the explicit dependence of $\boldsymbol{\delta r}(\boldsymbol{x})$ on the input image $\boldsymbol{x}$ to enhance readability.

## B.6 Linear encoder

We can further simplify the $\mathcal{P}$-VAE architecture by making the encoder also linear. Let $\boldsymbol{W} \in \mathbb{R}^{K \times M}$ denote the encoder's weight matrix, and assume an exponential link function mapping the input to residual firing rates, i.e., $\boldsymbol{\delta r} = \exp(\boldsymbol{Wx})$.

Starting from eq. (24), substituting $\log \boldsymbol{\delta r} = \boldsymbol{Wx}$, and rearranging terms yields the following loss function for the $\langle \texttt{lin}|\texttt{lin} \rangle$ $\mathcal{P}$-VAE:

$$\mathcal{L}_{\text{Lin-PVAE}} = \boldsymbol{\lambda}^T \boldsymbol{\Phi}^T \boldsymbol{\Phi \lambda} + \boldsymbol{\lambda}^T \text{diag}(\boldsymbol{\Phi}^T \boldsymbol{\Phi} - \beta \boldsymbol{I}) + \boldsymbol{\lambda}^T (\beta \boldsymbol{W} - 2\boldsymbol{\Phi}^T)\boldsymbol{x} + \beta \sum_{i=1}^{K} r_i + \boldsymbol{x}^T \boldsymbol{x}. \tag{25}$$

# C Architecture, training, and hyperparameter details

## C.1 Datasets: additional details

We consider three datasets in this paper. We tile up the van Hateren dataset of natural images [104] and CIFAR10 into $16 \times 16$ patches and apply whitening and contrast normalization using the code made available by Boutin et al. [105]. This operation results in the following total number of samples:

- **van Hateren**:    #train = 107,520,    #validation = 28,224,
- **CIFAR$_{16 \times 16}$**:    #train = 200,000,    #validation = 40,000.

We use the MNIST dataset primarily for the downstream classification task. After the training is done, we use the following train/validation split to evaluate the models:

- **K-nearest neighbor classification** (tables 4 and 6): For this task, we only make use of the validation set for both training and testing of the classifier. We divide up the $N = 10,000$ validation samples into two disjoint sets of $N = 5,000$ samples each. We then draw random samples (without replacement) from the first half and use them for training the KNN classifier. We then test the performance on the other half.

- **Shattering dimensionality** (tables 4 and 6, last column): We use the entire MNIST training set ($N = 60,000$ samples) to train logistic regression classifiers on extracted representations. We then test the results using the entire validation set ($N = 10,000$ samples).

## C.2 Architecture details

For sparse coding results, we focused on models with linear decoders. For the fully linear models (Figs. 4 and 10) both the encoder and decoder were linear layers, without bias.

For the convolutional components, we use residual layers without batch norm. For van Hateren and CIFAR$_{16 \times 16}$ datasets, the encoders had 5 layers ($2 \times$ conv each). The decoders had 8 convolutional layers ($1 \times$ conv each). For the MNIST dataset, the encoders had 7 layers ($2 \times$ conv each). The decoders had 10 convolutional layers ($1 \times$ conv each). For all convolutional encoders, the output from ResNet was followed by a learned pooling layer. The pooled output was then fed into a feed-forward layer inspired by Transformers [145], which includes a layer norm as the final operation, the output of which was fed into a linear layer that projects features into posterior distribution parameters. For all convolutional decoders, nearest neighbor upsampling was performed to scale up the spatial dimension of reconstructions, as suggested by Child [34].

We experimented with both leaky_relu and swish activation functions [146, 147], and found that swish consistently outperformed leaky_relu in all our experiments across datasets and VAE models.

Please see our code for the full architecture details.

## C.3 Training details

We used a variety of learning rates and batch sizes, depending on the dataset and architecture. For $\langle \texttt{lin}|\texttt{lin} \rangle$ and $\langle \texttt{conv}|\texttt{lin} \rangle$ models, we used $lr = 0.005$, and for $\langle \texttt{conv}|\texttt{conv} \rangle$ models we used $lr = 0.002$. All models were trained using the AdaMax optimizer [148] with a cosine learning rate schedule [149]. Please see our code for the full details of training hyperparameters. Overall, we trained 195 VAE models, $n = 5$ seeds each, resulting in a total of $195 \times 5 = 975$ VAEs. For sparse coding models, we ran ISTA [81, 82] and LCA [80] with 270 hyperparameter combinations each. Training all models took roughly a week on 8 RTX 6000 Ada GPUs.

**Temperature annealing for discrete VAEs.**    We also annealed the temperature from a large value to a smaller value during the same first half of training for $\mathcal{P}$-VAE and $\mathcal{C}$-VAE. We found that the specific functional form of temperature annealing (e.g., linear, exponential, etc.) did not matter as much as the final temperature (Fig. 9). For both $\mathcal{P}$-VAE and $\mathcal{C}$-VAE, we start from $T_{\text{start}} = 1.0$ and anneal down to $T_{\text{stop}} = 0.05$ for $\mathcal{P}$-VAE, and $T_{\text{stop}} = 0.1$ for $\mathcal{C}$-VAE. We found that the $\mathcal{C}$-VAE performance was not very sensitive to the choice of $T_{\text{stop}}$, corroborating previous reports [89, 98].

The $\mathcal{P}$-VAE was relatively more sensitive to the value of $T_{\text{stop}}$, and we found marginal improvements when reducing it from 0.1 to 0.05. See Fig. 9 for comprehensive experiments exploring the effect of the final temperature, as well as a "hard-forward" training method where we set $T = 0$ in the forward pass (ensuring integer samples) and use a non-zero $T$ only during the backward pass (surrogate gradients). We find that our "relaxed Poisson" approach (Fig. 3) consistently outperforms the hard-forward approach.

**KL annealing for VAEs.** For all VAE models, we annealed the KL term during the first half of the training, which is known to be an effective trick in training VAEs [2, 33, 142, 150, 151].

### C.3.1 Training: sparse coding models

To fit LCA and ISTA models, we explored a combination of 6 $\beta$ schedules (same $\beta$ as in eq. (1)), 3 numbers of iteration (for inference), 3 learning rates, and 5 different seeds (for dictionary initialization). The code for LCA was obtained from the public python library "lca-pytorch" ([152]), and the code for ISTA was obtained from public "sparsecoding" repository of the Redwood Center for Theoretical Neuroscience (with added clipping of coefficients to be nonnegative, following the thresholding step).

We explored learning rates of $1 \times 10^{-1}$, $1 \times 10^{-2}$, and $1 \times 10^{-3}$. We trained all models for 100 epochs. We scheduled the $\beta$ parameters linearly, starting from $\beta_{\text{start}}$, and stepped it up every five epochs by $\beta_{\text{step}}$, until it reached $\beta_{\text{end}}$. We explored the following $\beta$ schedules (expressed as $\beta_{\text{start}}$:$\beta_{\text{end}}$:$\beta_{\text{step}}$):

$$0.05{:}0.7{:}0.1, \quad 0.01{:}0.1{:}0.01, \quad 0.1{:}1.0{:}0.1, \quad 0.05{:}0.7{:}0.05, \quad 0.05{:}0.5{:}0.05, \quad 0.1{:}0.1{:}0$$

We also explored the inference iteration limits of 100, 500, and 900 iterations. We selected the best fits to include in the main results shown in Figs. 4 and 5.

## D  Supplementary results

In this section, we include additional results that further support those reported in the main paper, including:

Table 5 contains the negative ELBO values for all VAE models with a linear decoder. This table reveals a comparable performance between using Monte Carlo samples to estimate gradients, versus optimizing the exact loss (see eqs. (4), (21), (23) and (24)), highlighting the effectiveness of our Poisson reparameterization algorithm.

Figure 7 uses the same data from the main paper Table 2 to visualize the effects.

Figure 8 shows the dependence of loss on latent dimensionality. We find that increasing the number of latent dimensions consistently improves ELBO for $\langle$`conv`|`lin`$\rangle$ architectures, but $\langle$`lin`|`lin`$\rangle$ models either overfit (for van Hateren) or fail to improve (for CIFAR$_{16 \times 16}$) once $K$ becomes large.

Figure 9 demonstrates the robustness of our Poisson reparameterization trick (Algorithm 1) to variations in the temperature parameter. Importantly, we also explore a "hard-forward" training approach, where we fix $T = 0$ during the forward pass but allow $T > 0$ in the backward pass. This is also known as *surrogate gradients*. We find that, somewhat surprisingly, this hard-forward method performs significantly worse than our "relaxed Poisson" approach (Fig. 3).

Figure 10 shows how the distribution of KL values (or the norm of decoder weights in the case of linear decoders) can be used to determine dead neurons that don't contribute to the encoding of information.

Table 6 contains the full set of downstream classification results. Related to Table 4.

Figure 11 shows the performance of a simple linear classifier (logistic regression) trained on unsupervised representations learned by various $\langle$`conv`|`conv`$\rangle$ VAEs. We find that increasing the latent dimension ($K$) generally improves the performance of $\mathcal{P}$-VAE, but at lower dimensions, other methods like $\mathcal{L}$-VAE and $\mathcal{G}$-VAE can outperform it.

Figure 12 shows MNIST samples generated from the latent space of different $\langle$`conv`|`conv`$\rangle$ VAE models, as well as their reconstruction performance.

Table 5: The reparameterized gradient estimators work as well as exact ones, across datasets and encoder architectures (linear vs. conv). Note that exact gradients are only computable for linear decoders (see eqs. (21), (23) and (24)). The values are negative ELBO (lower is better), shown as mean$\pm$99% confidence interval calculated from $n = 5$ different random initializations. For MNIST, our use of Gaussian conditional likelihoods means the numerical performance values are not directly comparable to studies that use binarized MNIST with a cross-entropy decoder. EX, exact, MC, Monte-Carlo, ST, straight-through [107]. See also Table 2 and supplementary Fig. 7.

| Model | | van Hateren | | CIFAR$_{16\times16}$ | | MNIST | |
|---|---|---|---|---|---|---|---|
| | | $\langle$lin$\|$lin$\rangle$ | $\langle$conv$\|$lin$\rangle$ | $\langle$lin$\|$lin$\rangle$ | $\langle$conv$\|$lin$\rangle$ | $\langle$lin$\|$lin$\rangle$ | $\langle$conv$\|$lin$\rangle$ |
| | EX | $168.0_{\pm.8}$ | $162.4_{\pm.2}$ | $167.1_{\pm.2}$ | $162.1_{\pm.1}$ | $41.5_{\pm.1}$ | $39.7_{\pm.2}$ |
| $\mathcal{P}$-VAE | MC | $167.2_{\pm.1}$ | $163.4_{\pm.1}$ | $167.3_{\pm.1}$ | $162.9_{\pm.2}$ | $41.7_{\pm.2}$ | $40.1_{\pm.2}$ |
| | ST | $179.3_{\pm.1}$ | $179.4_{\pm.1}$ | $182.3_{\pm.1}$ | $182.3_{\pm.2}$ | $44.8_{\pm.1}$ | $44.2_{\pm.1}$ |
| $\mathcal{G}$-VAE | EX | $160.3_{\pm.1}$ | $154.4_{\pm.1}$ | $165.9_{\pm.1}$ | $149.2_{\pm.0}$ | $40.6_{\pm.1}$ | $40.0_{\pm.1}$ |
| | MC | $160.3_{\pm.1}$ | $154.4_{\pm.1}$ | $165.9_{\pm.1}$ | $149.2_{\pm.1}$ | $40.7_{\pm.1}$ | $40.1_{\pm.0}$ |
| | EX | $174.9_{\pm.1}$ | $186.3_{\pm.8}$ | $177.1_{\pm.1}$ | $180.6_{\pm.5}$ | $56.1_{\pm.7}$ | $59.1_{\pm.0}$ |
| $\mathcal{C}$-VAE | MC | $170.5_{\pm.1}$ | $171.9_{\pm.2}$ | $174.7_{\pm.1}$ | $176.5_{\pm.1}$ | $39.7_{\pm.2}$ | $59.1_{\pm.0}$ |
| | ST | $174.2_{\pm.2}$ | $181.1_{\pm.3}$ | $180.2_{\pm.0}$ | $185.6_{\pm.2}$ | $49.3_{\pm.1}$ | $63.8_{\pm3.4}$ |
| $\mathcal{L}$-VAE | EX | $167.3_{\pm.0}$ | $159.0_{\pm.2}$ | $170.1_{\pm.1}$ | $154.3_{\pm.1}$ | $42.1_{\pm.1}$ | $41.0_{\pm.0}$ |
| | MC | $167.3_{\pm.0}$ | $159.2_{\pm.2}$ | $170.1_{\pm.1}$ | $154.5_{\pm.1}$ | $42.1_{\pm.0}$ | $41.0_{\pm.0}$ |

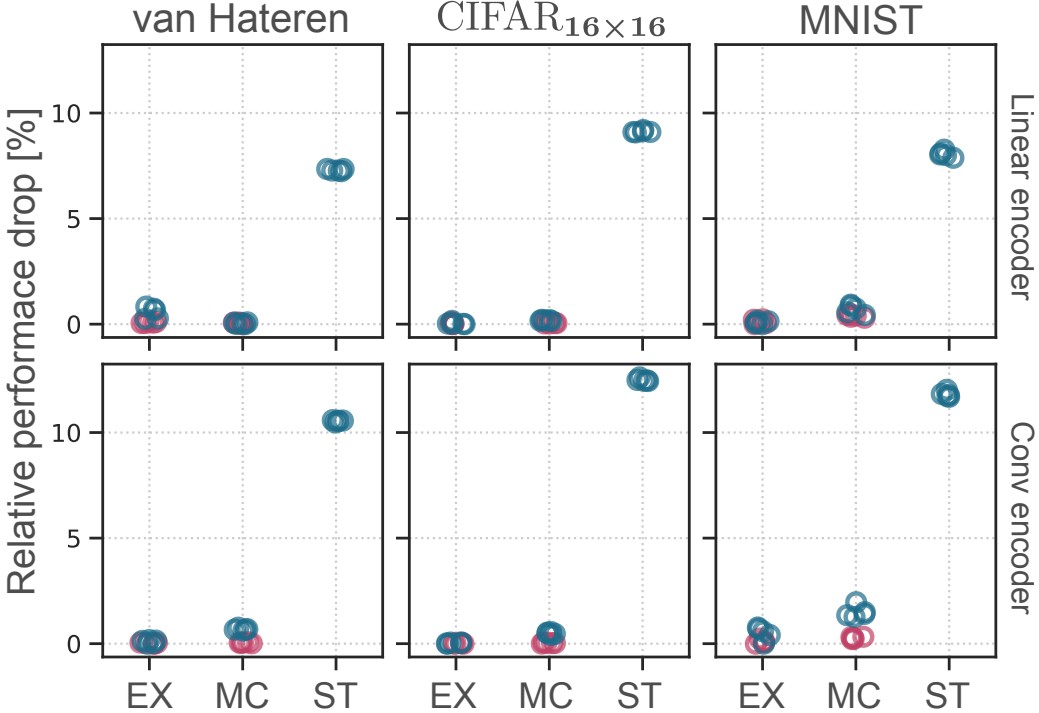

Figure 7: Performance drop relative to the best fit. Blue circles indicate $\mathcal{P}$-VAE results, red circles indicate $\mathcal{G}$-VAE results, and each set of $n = 5$ circles corresponds to five random initializations. Using Monte Carlo samples [153] and our Poisson reparameterization trick (Algorithm 1) to estimate gradients performs comparably to using exact gradients (see eqs. (21), (23) and (24)). Table 2 provides a tabular summary of these results. EX, exact, MC, Monte-Carlo, ST, straight-through [107].

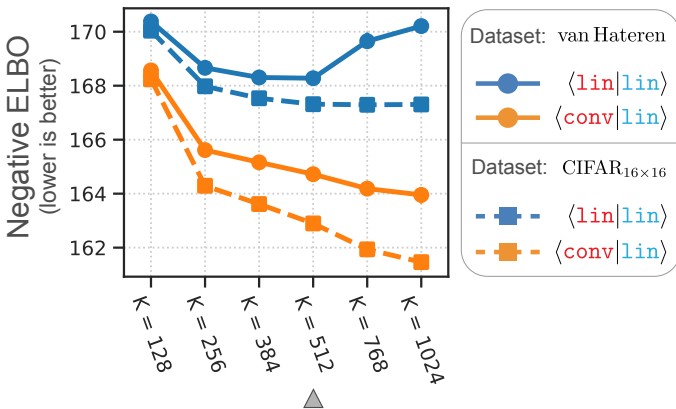

Figure 8: The effect of latent dimensionality on model performance across datasets and encoder architectures. For all convolutional encoder cases, ELBO improves as a function of latent dimensionality. However, for linear encoders, we see that the van Hateren dataset starts to overfit for $K > 512$, and it stagnates for the CIFAR$_{16 \times 16}$ dataset. In conclusion, more expressive encoders can find nonlinear features, represented using additional latent dimensions, but simple linear encoders struggle to utilize additional dimensions. The gray triangle indicates the setting used in the main results.

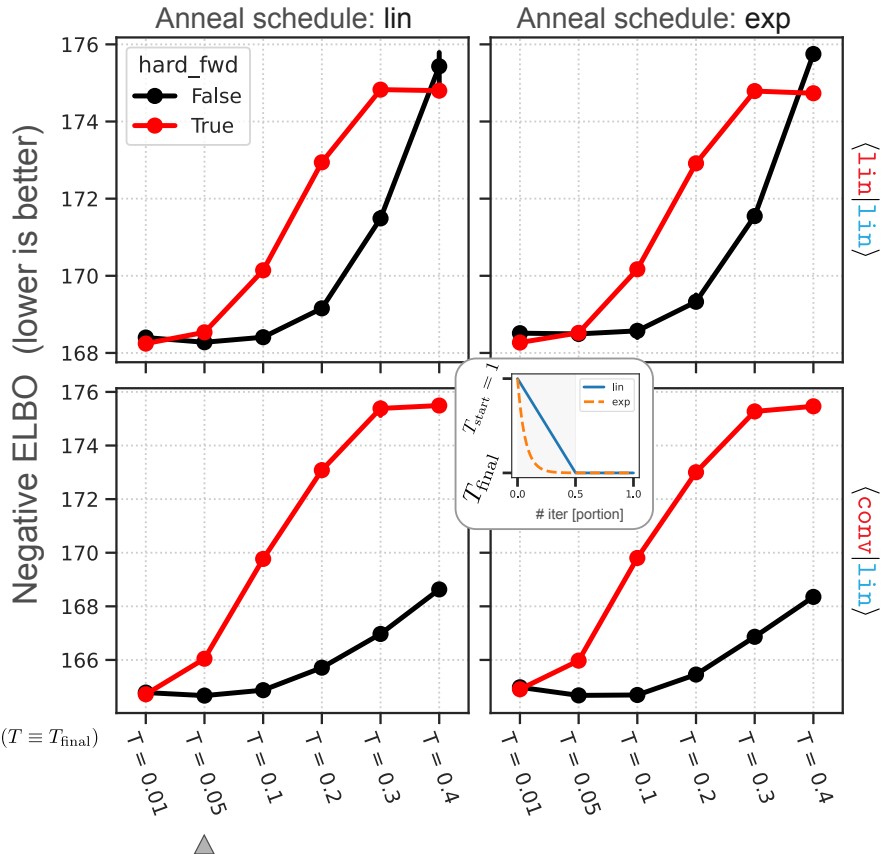

Figure 9: Performance as a function of the final temperature ($T_{\text{final}}$), annealing schedule (linear vs. exponential; inset), and the "hard-forward" approach. The hard-forward approach uses exact integer samples ($T = 0$) in the forward pass and applies nonzero temperatures only in the backward pass (i.e., "surrogate gradients"). Although all results are evaluated at $T = 0$ during testing, the hard-forward approach still underperforms our "relaxed Poisson" method (Fig. 3), which employs continuous (floating) samples during training due to a non-zero $T$ (Algorithm 1). The gray triangle indicates the setting used in the main results: $T_{\text{final}} = 0.05$ with a linear annealing schedule.

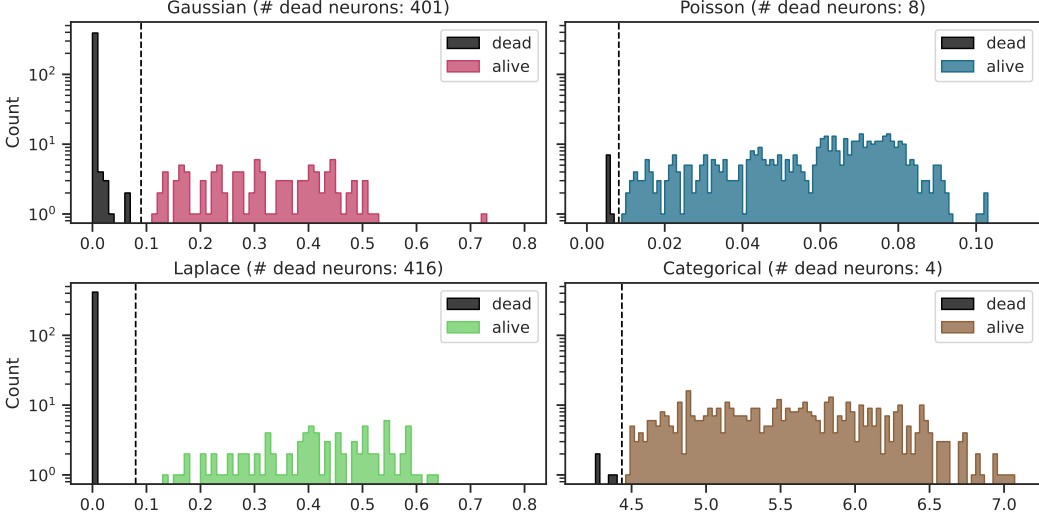

Figure 10: Identifying dead neurons using a histogram-based method. We bin the KL values and determine the gap between small values and larger ones. We identify neurons with KL values lower than the identified threshold (black dashed lines) and pronounce them dead. The figure shows the distribution of KL values over all neurons ($K = 512$) for $\mathcal{P}$-VAE, $\mathcal{G}$-VAE, and $\mathcal{L}$-VAE. The KL term is a single number for the $\mathcal{C}$-VAE because its latent space consists of a single one-hot categorical distribution with $K = 512$ categories. Therefore, for the $\mathcal{C}$-VAE, we use the distribution of decoder weight norms instead. These are the same models shown in Fig. 4, where both encoder and decoder are linear. Table 3 uses this method to quantify the proportion of active neurons for VAEs across different datasets and the choice of encoder architectures.

Table 6: Geometry of representations. Full set of results. Related to Table 4.

| Latent dim. | Model | KNN classification ($N$, # labeled samples) | | | Shattering dim. |
|---|---|---|---|---|---|
| | | $N = 200$ | $N = 1,000$ | $N = 5,000$ | |
| $K = 10$ | $\mathcal{P}$-VAE | **0.815**±.002 | **0.919**±.001 | **0.946**±.017 | **0.797**±.009 |
| | $\mathcal{C}$-VAE | 0.705±.002 | 0.800±.002 | 0.853±.040 | **0.795**±.006 |
| | $\mathcal{L}$-VAE | 0.757±.003 | 0.869±.002 | **0.924**±.028 | 0.751±.008 |
| | $\mathcal{G}$-VAE | 0.673±.003 | 0.813±.002 | 0.891±.033 | 0.758±.007 |
| | $\mathcal{G}$-VAE $_{+\mathrm{relu}}$ | 0.694±.003 | 0.817±.003 | 0.877±.045 | 0.762±.007 |
| | $\mathcal{G}$-VAE $_{+\mathrm{exp}}$ | 0.642±.003 | 0.784±.002 | 0.863±.032 | 0.737±.008 |
| $K = 50$ | $\mathcal{P}$-VAE | **0.825**±.002 | **0.927**±.001 | **0.957**±.005 | **0.935**±.003 |
| | $\mathcal{C}$-VAE | 0.770±.002 | 0.880±.001 | 0.920±.009 | 0.899±.004 |
| | $\mathcal{L}$-VAE | 0.710±.003 | 0.836±.003 | 0.902±.038 | 0.770±.007 |
| | $\mathcal{G}$-VAE | 0.604±.003 | 0.746±.002 | 0.837±.022 | 0.743±.007 |
| | $\mathcal{G}$-VAE $_{+\mathrm{relu}}$ | 0.710±.002 | 0.844±.002 | 0.904±.026 | 0.786±.006 |
| | $\mathcal{G}$-VAE $_{+\mathrm{exp}}$ | 0.694±.003 | 0.836±.002 | 0.906±.027 | 0.762±.007 |
| $K = 100$ | $\mathcal{P}$-VAE | **0.807**±.002 | **0.925**±.001 | **0.958**±.013 | **0.949**±.002 |
| | $\mathcal{C}$-VAE | 0.753±.002 | 0.876±.001 | 0.925±.005 | 0.884±.004 |
| | $\mathcal{L}$-VAE | 0.701±.004 | 0.830±.003 | **0.896**±.046 | 0.767±.007 |
| | $\mathcal{G}$-VAE | 0.636±.003 | 0.789±.002 | 0.875±.024 | 0.763±.007 |
| | $\mathcal{G}$-VAE $_{+\mathrm{relu}}$ | 0.757±.002 | 0.881±.001 | **0.933**±.019 | 0.818±.006 |
| | $\mathcal{G}$-VAE $_{+\mathrm{exp}}$ | 0.695±.003 | 0.846±.002 | 0.918±.024 | 0.793±.006 |

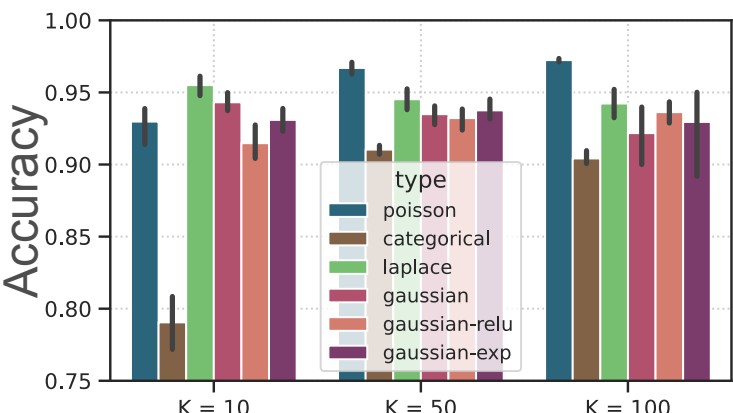

Figure 11: Downstream classification performance using a simple linear classifier. After unsupervised training of $\langle \text{conv}|\text{conv}\rangle$ VAEs on MNIST, we extracted latent representations and applied logistic regression. For $K = 100$, $\mathcal{P}$-VAE achieves the highest accuracy, while for $K = 10$, both $\mathcal{L}$-VAE and $\mathcal{G}$-VAE outperform it.

Figure 12: Generated samples (left) and reconstruction performance (right). These results shown here are from models with a $\langle \text{conv}|\text{conv}\rangle$ architectures and latent dimensionality of $K = 10$.

