# OpenReview forum: "Poisson Variational Autoencoder"
_NeurIPS.cc/2024/Conference — NeurIPS 2024 spotlight_

### Official Review · Reviewer_gEE3 · 2024-07-08

**Soundness:** 4
**Presentation:** 4
**Contribution:** 3
**Rating:** 7
**Confidence:** 4

**Summary:**

The paper proposes a variation on variational auto-encoders with a Poisson distribution over the latents. To make the model differentiable they use a differentiable sampling of the latent variables where the indicator function is replaced with a continuous approximation. They take inspiration from biological neural networks to develop the model and make connections to sparse coding to argue that their model can be used to understand sensory processing in the brain. Their experiments examine various key aspects of their model and compare to other significant VAE baselines. They provide evidence that their model learns sparse basis vectors for image datasets when compared to popular sparse coding algorithms and their model learns latent representations that perform better than baselines on downstream classification tasks.

**Strengths:**

- A Poisson variational auto-encoder is a somewhat novel though not as original contribution to the VAE zoo that which can clearly be situated among other VAE models. The connection to sparse coding and brain representations also well motivates the utility of the model for studying representations in the brain.
- The paper is very well written and was, for the most part, easy to follow. The model is well-explained and the figures served well to aid in understanding
- The experimental evaluation was quite thorough with the results providing strong evidence to support the paper's claims. The systematic analysis of various of the model's aspects (sparseness of representations and utility for downstream tasks) was well done and made clear to the reviewer what the capabilities and limitations of the model are.

**Weaknesses:**

- A significant weakness is a lack of study of the effect of the temperature parameter in the Poisson re-parameterization. How does this need to be set and what consequences for different temperatures? The z variables on line 6 of Algorithm 1 aren't integers and so how does  changing temperature affect the gradient? Appendix A.3 does talk about how the temperature is annealed during training but motivation for this approach isn't provided even though the temperature is a key aspect of their method.
- Also, a lack of discussion of the noise level that needs to be set for the likelihood $p(x | z)$ makes it unclear as to how the model can be re-configured to other datasets and what the consequences would be. What if we believed that the noise level in the data was much less than one (as is assumed with the Gaussian likelihood conditioned on the decoder output)

**Questions:**

- The outputs of the decoders in Fig. 1 should have noise added to them to illustrate that the likelihood isn't a delta distribution.
- Fig 1b is a bit confusing as the sampling of the latents isn't shown whereas it was shown in Fig 1a.
- Maybe Table 4 should be added to the main paper as it is the first experiment that is referred to and the reparameterization gradient is a crucial aspect of the model.
- Can a discussion be added on the slight differences between the P-VAE basis vectors and those of LCA and ISTA in Figure 2? Also what is the gold standard here? Is there a quantitative metric or can this only be determined qualitatively.

**Limitations:**

The authors have sufficiently discussed limitations.

---

> ### Author Rebuttal · Authors · 2024-08-07
>
> We thank the reviewer for their time and thoughtful comments.
>
> > A significant weakness is a lack of study of the effect of the temperature parameter in the Poisson re-parameterization.
>
> We agree. Please see our new rebuttal results, where we performed extensive experiments to address this point. We will include these new results in the paper.
>
> > The z variables on line 6 of Algorithm 1 aren't integers…
>
> Good point. This is true during training, but not during test time. We would like to emphasize that during validation, we always set $T = 0$, which results in integer samples drawn from a true Poisson distribution. We have included a discussion of this point and a figure in the global rebuttal and will integrate that into the final paper.
>
> Related to this point, in Fig. R1b we plot the distribution of samples at various temperatures, which shows the samples are indeed not integers for $T > 0$, but they approach integer values when $T \rightarrow 0$. Therefore, even during training, we can obtain "almost-integer" values if we anneal down to low temperatures. Interestingly, this does affect performance, and having these subtle non-integer values during training even improves model performance at test time with hard integers (Fig. R1a).
>
> > …how does changing temperature affect the gradient?
>
> We weren't sure exactly how to address this. We find that the model performance is robust for a range of values of $T_\mathrm{final} < 0.1$ (Fig. R1a), but we did not directly evaluate the effects on gradients. We are happy to address this more directly if you can suggest an analysis. Otherwise, we're inclined to interpret the performance results (Fig. R1a) instead of analyzing the gradients directly.
>
> > …a lack of discussion of the noise level that needs to be set for the likelihood $p(x \vert z)$...
>
> We construct our likelihood function, $p(x \vert z)  = \mathcal{N}(x; \mathrm{dec}(z), \sigma^2)$, by learning the Gaussian mean but using the same fixed variance for every pixel. Specifically, we chose $\sigma = 1/\sqrt{2}$, such that $-\log p(x \vert z) = ||x - \mathrm{dec}(z)||_2^2$. This choice of fixed variance is fairly standard in the VAE literature; however, others have drawn attention to the "disheartening" limitations of this approach ([Arvantidis et al., 2018](https://openreview.net/forum?id=SJzRZ-WCZ)), and we share their sentiment. We will add these comments to the paper to further clarify our choices.
>
> > The outputs of the decoders in Fig. 1 should have noise added to them to illustrate that the likelihood isn't a delta distribution.
>
> Thank you for pointing this out. We will edit the figure to highlight this.
>
> > Fig 1b is a bit confusing as the sampling of the latents isn't shown whereas it was shown in Fig 1a.
>
> Thank you for pointing this out. We will edit the figure to make it more consistent.
>
> > Maybe Table 4 should be added to the main paper…
>
> We plan to include a reduced version of Table 4 (or Fig. 4), along with Fig. R1b, in the main paper. The goal is to emphasize that, even though the approximate posterior is "relaxed Poisson," the final performance is almost as good as using exact gradients for training.
>
> > Can a discussion be added on the slight differences between the P-VAE basis vectors and those of LCA and ISTA in Figure 2? Also what is the gold standard here? Is there a quantitative metric or can this only be determined qualitatively.
>
> There is not really a gold standard here and, surprisingly, most of the sparse coding literature evaluates dictionaries qualitatively. One way to quantify the differences and similarities between pairs of dictionaries is by fitting parametric Gabor functions. These Gabor fits allow extracting and studying parameters such as orientation, spatial frequency, size, and location for each basis element. We briefly explored this direction for the rebuttal results. Please see Fig. R1e.
>
> We believe this line of inquiry deserves a more thorough investigation. We are happy to add a comparison of the distributions of Gabor fits to the different dictionaries if the reviewer thinks that would be helpful. However, we are inclined to leave this direction as future work.

---

> > ### Comment · Reviewer_gEE3 · 2024-08-10
> >
> > I would like to thank the authors for addressing my comments.
> >
> > The additions and changes proposed for the paper would certainly strengthen it and so I will raise my score to a 7

---

### Official Review · Reviewer_tb4S · 2024-07-10

**Soundness:** 3
**Presentation:** 3
**Contribution:** 3
**Rating:** 7
**Confidence:** 3

**Summary:**

This work introduces VAEs with Poisson-distributed latent variables and a Poisson reparameterization for efficient training. This approach has theoretical and empirical connections with sparse coding and behaves more similarly to biological networks that rely on discrete spike counts.

**Strengths:**

To the best of my knowledge, the method is original and well motivated. The paper is well written and generally clear. Though I cannot speak to the significance of this work from a neuroscience perspective, introducing Poisson latents is interesting in itself from a probabilistic models perspective. The experimental results are relatively extensive and there are several ablations.

**Weaknesses:**

- Posterior collapse in VAEs (in the sense of "some latent dimensions are not used") is not necessarily an issue per se. If this is an issue in the specific scenarios considered in this paper, I think this should be clarified.
- For the downstream classification tasks, I appreciate the experiments with different latent space sizes. However, using KNN as downstream classifier is a quite specific choice, and I would argue a simple linear probe is more common in the literature and seems like an intentionally missing baseline.
- While my expertise is in machine learning rather than neuroscience, I appreciate the value of ANNs that mimic biological networks. However, from an ML perspective, it would be helpful to discuss any potential challenges or limitations of this method (besides the limitations currently mentioned). The paper currently highlights the advantages of using Poisson latents in various aspects, which may seem overly optimistic. Since the goal is to bridge ANNs with biological systems rather than outperform benchmarks, a balanced discussion of both strengths and weaknesses would provide a more comprehensive perspective and would IMHO add value to this work.
- Since the main motivation is to learn brain-like representations, are there any datasets or experiments where this comparison with biological representations could actually be done? This might be the biggest missing part right now. If this is not feasible at all, it would also be fine to at least include a discussion for readers without a neuroscience background.
- More broadly, connections to neuroscience, as well as jargon, should be made more explicit/clear for pure ML people (e.g. "Gabor-like feature selectivity")
- Notation: at some point the authors start using $\boldsymbol{r}$ and $\boldsymbol{\delta}$ without introducing them in the main text (though the notation is better explained in Appendix). I would even recommend using $\boldsymbol{\delta}_r$ instead of $\boldsymbol{\delta r}$ which looks like a product, and is especially confusing when writing $\boldsymbol{r \delta r}$.

**Questions:**

See weaknesses.

**Limitations:**

Some limitations are addressed, but overall the presentation of the experimental results doesn't seem too balanced.

---

> ### Author Rebuttal · Authors · 2024-08-07
>
> We thank the reviewer for their time and thoughtful comments.
>
> > Posterior collapse in VAEs (in the sense of "some latent dimensions are not used") is not necessarily an issue per se. If this is an issue in the specific scenarios considered in this paper, I think this should be clarified.
>
> We agree that the concern over posterior collapse depends on the application and we will clarify this in the final text. Nevertheless, addressing the posterior collapse issue remains an active area of research ([He et al., 2019](https://openreview.net/forum?id=rylDfnCqF7), [Lucas et al., 2019](https://proceedings.neurips.cc/paper/2019/hash/7e3315fe390974fcf25e44a9445bd821-Abstract.html), [Razavi et al., 2019](https://openreview.net/forum?id=BJe0Gn0cY7), [Menon et al., 2022](https://openreview.net/forum?id=SrgIkwLjql9), just to name a few).
>
> In our review of existing literature at the intersection of VAEs and sparse coding, we found that posterior collapse has been a recurring issue there as well. Often, sparse coding results are evaluated based on what features are learned by the dictionary elements. For example, both [Csikor et al., 2023](https://www.biorxiv.org/content/10.1101/2023.11.29.569262v2) and [Geadah et al., 2024](https://www.biorxiv.org/content/10.1101/399246v3), used Laplace-distributed latents in VAEs, aiming to learn Gabor-like feature selectivity. However, they did not show the full set of dictionary elements. Our experiments revealed that ~80% of latent dimensions collapse for the L-VAE, resulting in noisy and therefore useless basis elements. This significantly deviates from the classical sparse coding results.
>
> One implication of our results for future work is that L-VAE should be avoided in favor of P-VAE if the goal is to develop a diverse set of dictionary elements that are reminiscent of classical sparse coding. We will add a discussion of why posterior collapse is relevant in our case to the final paper.
>
> > …using KNN as downstream classifier is a quite specific choice, and I would argue a simple linear probe is more common in the literature and seems like an intentionally missing baseline.
>
> We reasoned that KNN is a good choice for evaluating the learned representations because it is a non-parametric method and its performance is directly influenced by the geometry of representations—which is what we were interested in evaluating.
>
> For a more complete model evaluation, we performed simple logistic regression classification as part of the rebuttal results (Fig. R1c). We found that P-VAE achieves the best overall performance for latent dimensionally of $K = 100$. But for $K = 10$, both L-VAE and G-VAE outperform P-VAE. We plan to include these new results in Tables 3 and 5, alongside the KNN and shattering dim results.
>
> > …a balanced discussion of both strengths and weaknesses would provide a more comprehensive perspective and would IMHO add value to this work.
>
> We agree. However, we thought our presentation was fairly balanced. For example, we show in Fig. 6 that both the quality of generated samples, and reconstruction performance, are higher for continuous VAEs compared to discrete ones. We also highlight a large amortization gap that remains between P-VAE and sparse coding models. That said, within discrete models, P-VAE still performs better than C-VAE and there are advantages to P-VAE, which we highlight throughout the paper. We will make sure that our discussion of strengths and limitations is more clear in the final paper.
>
> > Since the main motivation is to learn brain-like representations, are there any datasets or experiments where this comparison with biological representations could actually be done? This might be the biggest missing part right now.
>
> We agree that a comparison with biological representations is an obvious next step and there are several datasets available for this. In the present work, we demonstrated the connection between Poisson VAEs and sparse coding and evaluated the representations learned by continuous and discrete VAEs. Previous work has evaluated VAEs for predicting neural activity ([Vafaii et al., 2023](https://openreview.net/forum?id=1wOkHN9JK8)), and has found that benchmarking, as is standardly done, does not discriminate well between different models. A more thorough evaluation of the learned representations is necessary and there isn’t room here to do that justice, although, this is an immediate plan of ours. Since both neurons in the brains and P-VAE encode information in firing rates, we believe the P-VAE will learn brain-like representations.
>
> We have discussed some applications in the global rebuttal and will integrate some of the points made here into the discussion in the final paper.
>
> > …connections to neuroscience, as well as jargon, should be made more explicit/clear for pure ML people…
>
> Thank you for pointing this out. We will amend the language to enhance clarity.
>
> > Notation: at some point the authors start using 𝑟 and 𝛿 without introducing them in the main text…
>
> We have $\log\delta r \in \mathbb{R}^{K}$, such that $\delta r \in \mathbb{R}_+^{K}$. Additionally, by $r \delta r$, we mean $r \odot \delta r$, where $\odot$ is the Hadamard or element-wise multiplication. We will clarify this in the text.

---

> > ### Comment · Reviewer_tb4S · 2024-08-10
> >
> > Thank you for your thorough rebuttal and for carefully addressing all the concerns raised by myself and the other reviewers. If the final version of the paper incorporates the promised revisions, I believe it will be a strong contribution that merits acceptance.
> >
> > I have updated my score from 5 to 7.

---

### Official Review · Reviewer_w8Jn · 2024-07-11

**Soundness:** 4
**Presentation:** 4
**Contribution:** 3
**Rating:** 7
**Confidence:** 3

**Summary:**

Inspired by biological neurons, a new type of variational autoencoder, the Poisson variational autoencoder ($\mathcal{P}$-VAE) is proposed. The $\mathcal{P}$-VAE uses discrete latent states with Poisson priors, and learns sparse discrete representations of the data similar to sparse coding methods. The authors compare their method with classical sparse coding algorithms and obtain learned representations that are less prone to posterior collapse and are better for downstream classification tasks.

**Strengths:**

- This paper is very well-written and ideas are presented very clearly. The writing is clear and concise with little to no typos. The contributions are outlined and emphasized throughout the paper. Background to the work are explained thoroughly. The figure and algorithm box serve to communicate the main algorithm clearly to the reader. The color coding of inference and generative components in section 3 makes it easy to parse the equations and grok the main $\mathcal{P}$-VAE algorithm.
- The main arguments of the paper are supported well by experiments. The experiment section presents extensive analysis on the representation learning capabilities of the $\mathcal{P}$-VAE, and compared it to many existing VAE models, including both continuous and discrete ones. In-depth discussions are made with respect to the Gabor-like quality of learned filters, avoiding posterior collapse, sparsity and effectiveness in downstream tasks.

**Weaknesses:**

My main concern for this work is in its limited impact. While to my knowledge it is true that applying the Poisson prior to discrete VAEs as a sparsity-inducing constraint is novel, I fail to see how this is fundamentally different that other discrete VAEs with regularization. Further to this point, the comparisons to VAE with the concrete distribution ($\mathcal{C}$-VAE) seems relatively weak, as the latter has less dead neurons and also seems to achieve more sparsity for the same reconstruction performance (Figure 3a). The authors also commented that LCA models drastically outperform the $\mathcal{P}$-VAE with the convolution encoder, making it questionable if one should use $\mathcal{P}$-VAE for its sparsity.

To the authors’ credit, the paper does state that part of the appeal of the $\mathcal{P}$-VAE is in its biological plausibility. Unfortunately I don’t think this point is expanded upon to a satisfactory degree in the text, leaving much to be desired. Personally I would love to see more technical discussion on the biological plausibility of the $\mathcal{P}$-VAE, and concrete examples on the types of tasks and inquiries it unlocks for neuroscientific studies that similar models cannot solve adequately.

Despite the above points, I still think that this is a solid paper with potential contributions to the computational neuroscience community.

**Questions:**

- I am confused by the statement on lines 56-57: “facilitating linear separability of categories in a downstream classification task with a much better (5x) sample efficiency.” Isn’t it true that the increase in sample efficiency is for KNN classification but not in the linear separability? In that case this sentence is misleading.
- What’s the significance of the global prior parameter $r$? Have you tried ablation studies where the posteriors are directly parameterized?
- What’s the takeaway for the “$\mathcal{P}$-VAE learns sparse representations” section? How should we view the amortization gap you mentioned on line 277?
- There are multiple bold entries for some columns in tables 3 and 5, what do these mean?
- Can you talk more about where you believe the $\mathcal{P}$-VAE model should be applied and how it can be impactful to the scientific community?

**Limitations:**

The authors have adequately addressed the limitations of their work.

---

> ### Author Rebuttal · Authors · 2024-08-07
>
> We thank the reviewer for their time and thoughtful comments.
>
> > My main concern for this work is in its limited impact.
>
> We hope that we have addressed this concern in the general rebuttal. We developed P-VAE with neuroscience applications in mind, where we feel it will have a big impact. However, there are several other exciting future directions that we have discussed in the global rebuttal, which we will expand upon and include in the final paper.
>
> > …the comparisons to VAE with the concrete distribution (𝐶-VAE) seems relatively weak, as the latter has less dead neurons and also seems to achieve more sparsity for the same reconstruction performance.
>
> C-VAE is hard to compare to for a number of reasons which we'll discuss here. But first, we want to emphasize the P-VAE with a convolutional encoder has much better reconstruction performance than C-VAE, with a comparable sparsity level. That said, the sparsity comparison is hard to make because C-VAE is a one-hot categorical distribution, which means it has a fixed sparsity of $1-1/K$ (~99% when $K=512$). P-VAE can sweep out a tradeoff between sparsity and reconstruction (something akin to a rate-distortion curve). In contrast, the sparsity of C-VAE is solely determined by the latent dimensionality.
>
> Further, Because C-VAE has a single KL value, the comparison between dead neurons is also not straightforward, which is why we excluded C-VAE from Table 2. For the C-VAE, we used the norm of basis elements as a measure of the dead neurons. We explain this in the caption of Fig. 5.
>
> We will add text to discuss these points more explicitly in the final paper.
>
> > …The authors also commented that LCA models drastically outperform the 𝑃-VAE with the convolution encoder, making it questionable if one should use 𝑃-VAE for its sparsity.
>
> We were surprised to see such a strong performance for LCA, a "shallow" model from 2008. We speculate this is because LCA performs iterative inference, whereas P-VAE, in its current form, uses a single forward pass for inference (amortized inference). This observation has inspired us to develop an iterative version of the P-VAE in subsequent work, which we hope will reduce the amortization gap and either match or surpass LCA performance.
>
> That said, it is important to note that LCA necessarily has to work with a linear generative model ([Rozell et al., 2008](https://direct.mit.edu/neco/article-abstract/20/10/2526/7343/Sparse-Coding-via-Thresholding-and-Local?redirectedFrom=fulltext)), whereas P-VAE (like other VAEs) could have a nonlinear (deep) decoder. We primarily focused on linear decoders in this paper because of the close connection between the P-VAE loss and sparse coding. Two future directions could be to explore P-VAEs with more typical deep decoders or, as we mentioned above, with iterative inference, which we believe is a promising future direction for closing the amortization gap in VAEs more generally.
>
> > Personally I would love to see more technical discussion on the biological plausibility of the 𝑃-VAE, and concrete examples on the types of tasks and inquiries it unlocks for neuroscientific studies that similar models cannot solve adequately.
>
> Thank you for this comment. We have included a discussion on neuroscience applications in the global rebuttal which attempts to address this point. We will include more discussion on this point in the final version of the paper.
>
> > I am confused by the statement on lines 56-57: “facilitating linear separability of categories in a downstream classification task with a much better (5x) sample efficiency.” Isn’t it true that the increase in sample efficiency is for KNN classification but not in the linear separability? In that case this sentence is misleading.
>
> You are correct. That sentence should be broken into two separate points. We will clarify that the linear separability claim is supported by the shattering dim results, while the sample efficiency claim is supported by the KNN results.
>
> > What’s the significance of the global prior parameter 𝑟? Have you tried ablation studies where the posteriors are directly parameterized?
>
> We did not consider an independently parameterized posterior, because our current implementation has two desirable features:
>
> 1. a direct connection to predictive coding; and,
> 2. a nice factorization of the KL term.
>
> However, this question led us to investigate the learned global prior parameters, $r$, for which we thank the reviewer. We found that the prior rates learn an efficient representation of the natural scene statistics. We describe this exciting result in the global rebuttal (Fig. R1e).
>
> > What’s the takeaway for the "𝑃-VAE learns sparse representations" section? How should we view the amortization gap you mentioned on line 277?
>
> There are two main takeaways. First, P-VAE produces sparser representations than continuous VAE counterparts, with a comparable reconstruction performance. Second, there remains a substantial amortization gap between P-VAE (which uses amortized inference) and true sparse coding, which employs iterative inference. In our experiments, LCA converged in typically hundreds of iterations, whereas P-VAE does inference in one shot. Developing an iterative P-VAE and using it to close the amortization gap is our top priority for future work.
>
> > There are multiple bold entries for some columns in tables 3 and 5, what do these mean?
>
> Bold indicates values that passed a significance threshold using statistical tests. We will clarify in the final paper that the bolded models perform similarly, such that their performance is statistically indistinguishable.
>
> > Can you talk more about where you believe the 𝑃-VAE model should be applied and how it can be impactful to the scientific community?
>
> Yes. Multiple reviewers have requested this. We hope our global rebuttal addresses this adequately and will include a discussion on the significance of the work in the final paper.

---

> > ### Comment · Reviewer_w8Jn · 2024-08-14
> >
> > Thank you for your clarifications. I think my main concerns are well addressed and I now better understand why this work is important and interesting. I am hereby raising my score to a 7.

---

### Official Review · Reviewer_2Wt3 · 2024-07-14

**Soundness:** 4
**Presentation:** 4
**Contribution:** 3
**Rating:** 7
**Confidence:** 3

**Summary:**

In this paper, the authors propose a VAE model, the Poisson-VAE ($\mathcal{P}$-VAE), with a Poisson distributed prior and approximate posterior such that it works with Poisson distributed latents and demonstrate its sparse coding abilities on the van Hateren dataset and its efficacy on downstream classification tasks using the MNIST dataset.  The authors also present a Poisson reparametrization trick.

**Strengths:**

- The authors combine predictive coding and Poisson-distributed latents to obtain a neat P-VAE objective emphasizing sparsity without additional design constraints-- hence, elegantly buying amortized sparse coding.
- The authors also propose a new reparametrization trick, the Poisson reparametrization trick, which can potentially be more generally applicable.

Writing:
- The writing and presentation of the paper is very clear, making the derivation and rest of the math easily approachable.

Results:
- The authors seem to get very good results in terms of the values of the ELBOs obtained on MNIST as compared to pre-existing work. They also showcase linear separability of learnt features on MNIST.

**Weaknesses:**

These questions/remarks might also have arisen due to my lack of proper understanding, so I am willing to increase my score if these can be clarified:

- How do we know that the approximate posterior is Poisson?
- The temperature is controlling the sharpness of the thresholding-- what is the temperature being considered generally? How do we know that the cdf = 0.99999 across all these temperatures?

Experiments:
- How does the ELBO vary with the dimension of the overcomplete latent space considered?
- Is the VQ-VAE a valid baseline? If so, it would be helpful if the authors can mention why it hasn't been considered as a baseline in the paper.

**Questions:**

- The authors list this as a limitation of their work later, but is it possible to discuss potential reasons for a large amortization gap of the P-VAE as compared to the LCA/ISTA?

**Limitations:**

The authors adequately discuss limitations of their work, and its potential societal impact.

I would like to note another limitation of their work as not very illustrative experiments on downstream classification tasks (MNIST and CIFAR have been considered in the paper). The paper could benefit from experiments showing greater impact of sparse coding.

---

> ### Author Rebuttal · Authors · 2024-08-07
>
> We thank the reviewer for their time and thoughtful comments. We are also glad the reviewer finds our results "neat" and "elegant"!
>
> > How do we know that the approximate posterior is Poisson?
>
> During training, we use non-zero temperatures in our reparameterization algorithm. As a result, during training, the approximate posterior is a relaxed approximation to Poisson. In Fig. R1b we show samples drawn from our "relaxed Poisson" distribution at different temperatures. As $T \rightarrow 0$, the sampled distribution converges toward the true Poisson distribution at $T = 0$.
>
> However, we want to emphasize that we used relaxed Poisson only during training. At validation time, we always set $T = 0$. Therefore, the approximate posterior is exactly Poisson at test time.
>
> > …what is the temperature being considered generally?
>
> During the first half of training, we anneal temperatures from a large initial value,  such as $T_\mathrm{start} = 1.0$, to a small final value, $T_\mathrm{final}$. In the paper, as well as Figs. R1c and d, we report results obtained using $T_\mathrm{final} = 0.05$. Our new extensive experiments (Fig R1a) suggest this choice was reasonable.
>
> > How do we know that the cdf = 0.99999 across all these temperatures?
>
> Thank you for this interesting question, which we overlooked before. We determine ```n_exp``` in Algorithm 1 using the largest posterior rate in a given batch, $r_\mathrm{max}$. We observed that the distribution of rates in a batch is typically skewed and long-tailed. Consequently, the vast majority of the rates are much lower than $r_\mathrm{max}$. Therefore, this is a very conservative way of choosing ```n_exp```, and even if the "cdf = 0.99999" condition is not met for certain temperatures, it will be so only for a vanishingly small subset of rates.
>
> With that said, we explored this question empirically for temperatures encountered during training, using a few reasonable rate values. We found that this condition holds regardless of temperature (even more strongly for non-zero temperatures when the rate is small).
>
> > How does the ELBO vary with the dimension of the overcomplete latent space considered?
>
> We investigated this across datasets (van Hateren and CIFAR), and encoder architectures (linear versus convolutional). We report the results in Fig. R1d. We found that for all convolutional encoder cases, ELBO improves as a function of latent dimensionality. However, for linear encoders, we observed that the van Hateren dataset started to overfit for $K > 512$, and it stagnated for the $\mathrm{CIFAR}_{16 \times 16}$ dataset. In conclusion, more expressive encoders can find nonlinear features, represented using additional latent dimensions, but simple linear encoders struggle to utilize additional dimensions.
>
> > Is the VQ-VAE a valid baseline?
>
> We did not consider VQ-VAE as a baseline, because VQ-VAE does not optimize the ELBO loss, and thus, it is not technically a VAE. We think the naming is unfortunate and we will discuss this point in the final paper.
>
> > …discuss potential reasons for a large amortization gap of the P-VAE as compared to the LCA/ISTA?
>
> We suspect both LCA and ISTA perform well because they are iterative algorithms. Our P-VAE, in its current format, uses a single forward pass to perform inference. Prior work by [Marino and colleagues](https://proceedings.mlr.press/v80/marino18a.html) has shown that iterative inference can significantly decrease the amortization gap. As a future work, we are interested in developing iterative versions of P-VAE, which we hope will close the amortization gap and beat the best LCA and ISTA fits.
>
> > …not very illustrative experiments on downstream classification tasks…
>
> We agree with the reviewer here. We primarily used these downstream tasks to assess whether the geometry of representations is quantitatively different between P-VAE and alternative models. Our limited results in this area suggest they are different, which we plan to explore more rigorously later. The point of the current experiments was to establish there is indeed a difference in the geometry of representations, rather than fully exploring the differences and similarities, or highlighting particular applications of P-VAE.
>
> > The paper could benefit from experiments showing greater impact of sparse coding.
>
> We agree that a limitation of our paper is that we did not highlight applications where a P-VAE would have a greater impact. As described in the general rebuttal, we will discuss several potential future directions in the final paper, and we believe these warrant more attention than we have room for in this paper.
>
> We developed P-VAE with neuroscience applications in mind and one of the major advantages over other VAEs is that the latents can now be interpreted as neurons. That said, there are applications where sparse coding is applied, such as image processing and computational imaging, where P-VAE might shine. We have included a few of these potential directions in our general rebuttal, but still feel that they would require substantial experimental evaluation and should be left to future work.

---

> > ### Comment · Reviewer_2Wt3 · 2024-08-14
> > **reply to authors**
> >
> > I thank the authors for addressing my questions and concerns. Given the lack of more illustrative experiments (in my opinion), I maintain my fairly very positive score for the paper.

---

### Author Rebuttal · Authors · 2024-08-07

We thank the reviewers for their time and insightful feedback. We believe addressing the reviewers' comments will substantially improve our paper. We plan to include these changes in the form of two major components: **(i)** further discussion of the work significance; and, **(ii)** additional results, reported as a figure here (Fig. R1).

# (i) Work significance and contributions
## Utility for neuroscience
The neuroscience community increasingly uses ANNs to understand what biological neurons are selective for and why. ANNs have several advantages over real brains: all connections and activations are available to the investigator, in silico perturbations are inexpensive, and they do not require the use of animals.

Like biological neurons, the P-VAE generates spikes; therefore, its latents can be treated like neurons. This offers advantages over continuous unconstrained models such as L-VAE or G-VAE. Here, we focus on two examples, leaving a more comprehensive discussion to the final paper.

- **Example 1:** P-VAE supports direct comparison to causal perturbations in brains. Perturbation experiments selectively stimulate or silence neurons to assess the causal role of a group of neurons on perception. These types of experiments cannot be trivially compared to unconstrained VAEs, for which "stimulation" is complicated by the fact that the latents are signed. In contrast, the P-VAE can be readily used for designing and conducting in silico perturbation experiments, enabling an exciting potential transfer of insights to the in vivo setting.
- **Example 2:** "Maximally Exciting Inputs" (MEI; [Walker et al., 2019](https://www.nature.com/articles/s41593-019-0517-x)), which manipulate inputs following the gradients of a feed-forward network fit to biological neurons, have been used to understand what biological neurons are selective for. The concept of MEI requires that neuron activations be characterized by being more or less "excitable." Once again, this concept is readily applicable to P-VAE latents, but not the unconstrained VAEs.

There are just two examples out of many. We will expand upon this point in the final paper, mentioning more concrete examples of the application potential of the P-VAE and how it can help advance neuroscience research.

## Mechanistic interpretability
Sparse autoencoders (SAEs) have become popular for mechanistic interpretability ([Anthropic](https://transformer-circuits.pub/2023/monosemantic-features), [OpenAI](https://arxiv.org/abs/2406.04093)). Although not an initial motivation for our work, we have inadvertently built a probabilistic version of SAEs. We're excited about a hierarchical extension of P-VAE applied to both images and LLM activations to test if this approach extracts hierarchically organized semantic concepts.

## Reparameterization trick
We hope our reparameterization trick finds applications beyond VAEs, for example, in spiking neural networks (SNN). One of our new results shows that the "surrogate gradients" method—utilized heavily in the SNN literature—may be improved by relaxing the hard forward during training (see below).

## Hardware implementation
A key advantage of our architecture is its ability to learn discrete, sparse representations. The integer P-VAE representations eliminate the need for post hoc quantization, which is crucial for hardware implementation of models with float activations. This sparsity enhances memory efficiency and lowers energy use, highlighting P-VAE’s potential as a vision model implemented directly on hardware for robotics.

# (ii) New results (Figure R1)
## Temperature and performance (Fig. R1a)
Motivated by comments from Rev. gEE3, we performed additional experiments to quantify the effect of temperature (T) on the final model performance. Following standard practice ([Jang et al., 2017](https://openreview.net/forum?id=rkE3y85ee)), we annealed T from a large value ($T_\mathrm{start} = 1.0$) to a small value ($T_\mathrm{final} = 0.05$ in the main paper) during the first half of training. In Fig. R1a, we explore the effect of changing $T_\mathrm{final}$ on the van Hateren dataset, using two architectures (linear vs. convolutional encoders, linear decoder), and two annealing schedules (linear vs. exponential; Fig. R1e inset). We find values of $T_\mathrm{final} \leq 0.1$ work well, and both annealing schedules work well.

Importantly, all results were obtained using $T = 0$ during test time. We also experimented with the option of using a "hard forward" training scheme once the annealing is done (i.e., the last half of training), where we use non-zero temperatures only for the backward pass. This practice is known as "surrogate gradients." Somewhat surprisingly, we found that the surrogate gradients severely underperformed our "relaxed Poisson" approach. We anticipate this result will be highly interesting to the spiking neural network community, who rely mostly on surrogate gradients to train their networks. We plan to include this figure in the appendix and highlight the main takeaways in the main text.

## Natural image statistics learned in the prior rates (Fig. R1e)
Thanks to comments from Revs. gEE3 and w8Jn, we examined the properties of the P-VAE learned dictionary elements in conjunction with the global prior rates, r. See Fig. 1e and its caption.

We found that P-VAE prior rates are consistent with principles of efficient coding, which states brains should assign minimal neural resources to statistically dominant elements of the natural environment. Specifically, we found prior rates were lower for *cardinal* orientations, which are more common in natural image patches. This result mirrors biological brains and is another demonstration of the potential of P-VAE. We plan to explore this research direction more systematically in future work.

# Conclusion
P-VAE has shown promising results, opening up many exciting venues for future exploration at the intersection of machine learning and neuroscience.

---

### Decision · Program_Chairs · 2024-09-25

**Decision:**

Accept (spotlight)

**Comment:**

There is very strong consensus among the reviewers that this paper should be accepted to NeurIPS. The Poisson VAE represents a reasonably novel model, and is buttressed by both the biological plausibility of the model and a strong experimental section. Ideas from this paper appear to be generalizable to other domains, further increasing the impact.

The authors also provided a very strong rebuttal to the various points of contention from the reviewers, eventually resulting in the clear consensus of acceptance.